# High resolution ancient sedimentary DNA shows that alpine plant diversity is associated with human land use and climate change

Sandra Garcés-Pastor [1] ✉, Eric Coissac [2], Sébastien Lavergne[2], Christoph Schwörer [3], Jean-Paul Theurillat [4], Peter D. Heintzman[1], Owen S. Wangensteen [5,6], Willy Tinner[3], Fabian Rey[7], Martina Heer[7], Astrid Rutzer[7], Kevin Walsh [8], Youri Lammers [1], Antony G. Brown[1], Tomasz Goslar [9], Dilli P. Rijal [1], Dirk N. Karger [10], Loïc Pellissier [11,12], The PhyloAlps Consortium*, Oliver Heiri [7,29] & Inger Greve Alsos [1,29]

The European Alps are highly rich in species, but their future may be threatened by ongoing changes in human land use and climate. Here, we reconstructed vegetation, temperature, human impact and livestock over the past ~12,000 years from Lake Sulsseewli, based on sedimentary ancient plant and mammal DNA, pollen, spores, chironomids, and microcharcoal. We assembled a highly-complete local DNA reference library (PhyloAlps, 3923 plant taxa), and used this to obtain an exceptionally rich *sed*aDNA record of 366 plant taxa. Vegetation mainly responded to climate during the early Holocene, while human activity had an additional influence on vegetation from 6 ka onwards. Land-use shifted from episodic grazing during the Neolithic and Bronze Age to agropastoralism in the Middle Ages. Associated human deforestation allowed the coexistence of plant species typically found at different elevational belts, leading to levels of plant richness that characterise the current high diversity of this region. Our findings indicate a positive association between low intensity agropastoral activities and precipitation with the maintenance of the unique subalpine and alpine plant diversity of the European Alps.

Changing environmental conditions are displacing organisms out of their ranges, causing severe threats to biodiversity[1]. In high mountains, vegetation composition is naturally determined by temperature, moisture, landforms, and geomorphological processes along elevational belts. In addition, many mountain ranges have been under human influence for millennia[2], and both natural and anthropogenic forces have shaped the diversity of mountain vegetation[3]. Climate-based projections indicate an expected upward displacement of vegetation that will reduce habitat for alpine species, while changes in mountain land use with land abandonment in the last century have been causing increasing forest cover and loss of habitat for meadow species[3,4]. Plant remains in lake sediments allow us to explore vegetation responses to past climate changes and human activity, particularly at long time scales relevant for anticipating future vegetation responses to global changes. Therefore, detailed palaeoecological records representing the full range of plant types and functional

A full list of affiliations appears at the end of the paper. *A list of authors and their affiliations appears at the end of the paper.
✉ e-mail: sandra.garces-pastor@uit.no

groups that compose alpine and subalpine vegetation are needed to understand how long-term interactions of climate and humans affect overall biodiversity and survival of high elevation plants. However, some ecologically relevant groups such as graminoids and forbs are poorly represented in conventional palaeoecological records due to limited taxonomic resolution and low pollen production, respectively[5]. Recent advances in sedimentary ancient DNA (*sed*aDNA) have greatly improved our ability to give detailed insight into past diversity changes[6–8].

The European Alps are an important plant biodiversity hotspot[9], with ~4000 native plant species[10] distributed from the warm lowland Colline vegetation belt to the cold alpine Nival belt[10], which results from a complex interplay of natural factors over geological time-scales. Changes in climatic and environmental conditions in alpine regions brought suitable conditions for plant migration and speciation, resulting in the formation of high numbers of endemics[11,12]. However, human activities over millennia have modified, favoured and helped to maintain this diversity with the creation of new habitats at the local scale[13]. Humans have modified the subalpine and alpine landscapes since the Mesolithic, ca. 10 ka (1 ka = 1000 yr ago), by clearing small areas of forest to attract prey for hunting[14,15] while the introduction of agropastoral activities during the Neolithic (from 7.5 ka) drove a downward shift of the treelines[2,14–16]. Human-environment interactions in forested and open vegetation types such as the Subalpine zone led to a mosaic of different habitats that include species-rich meadows[8,10,15]. As a result, future changes in land use might imply a reduction in vegetation diversity of subalpine and alpine landscapes[17]. For example, the abandonment of high-mountain practices during the last half-century has reduced the plant diversity of subalpine pastures in many mountain ranges such

as the European Alps[13,18], the Pyrenees[19], and the Himalayas[20]. Understanding the contribution of past climate and land use in shaping alpine vegetation can help anticipate future impacts of global changes and may offer mitigation solutions.

Here, we reconstruct the vegetation around Lake Sulsseewli, located in the northern Swiss Alps (Fig. 1), with the aim to better understand the drivers of the species composition and elevational vegetation belts over the past 12,000 years. For this aim, we used a multiproxy approach consisting of plant *sed*aDNA, pollen, fossil chironomids for summer temperature reconstruction, precipitation data from CHELSA-TraCE21k model[21], geochemical proxies, and multiple independent indicators of human activity, that included microscopic charcoal (reflecting fire activity) and grazing indicators (coprophilous fungi spores and mammalian *sed*aDNA). Based on this comprehensive dataset, we investigated to what extent the current plant richness at Sulsseewli was related to past variations in climate and human activities. To achieve these aims, we assembled *trn*L P6 loop locus data from a new comprehensive taxonomic DNA reference database consisting of 3923 plant species collected in the Alps and 417 from the Carpathians (the PhyloAlps database; http://phyloalps.org/). The exceptionally high taxonomic resolution of the plant *sed*aDNA data allowed us to reconstruct both long-term changes in plant diversity and changes in plants that are particularly temperature-sensitive (i.e., with restricted elevational distributions), or are considered pastoral and arable indicators[22]. Our results suggest that vegetation was mainly driven by climate during the first half of the Holocene (~11 to 6 ka) and that the rise of diversity that characterises the present subalpine and alpine diversity was associated with extensive human pressure and increased precipitation from the Neolithic onwards, favouring the coexistence of taxa generally found in different vegetation belts. We

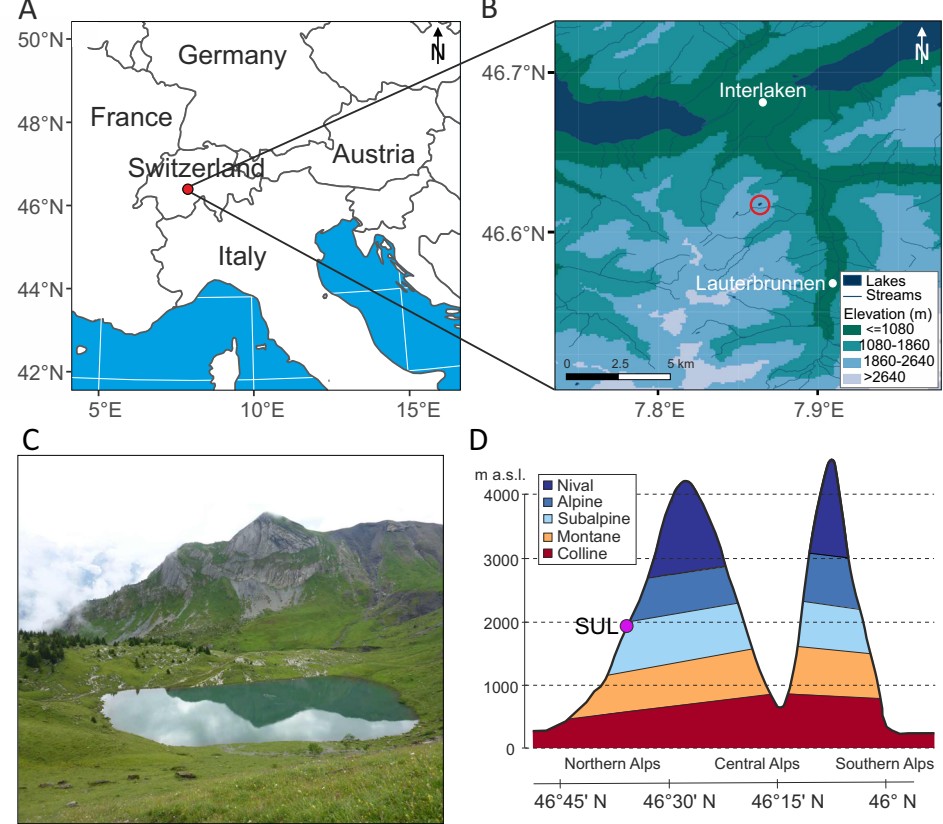

**Fig. 1 | Location of lake Sulsseewli, Switzerland. A** Map of Europe and a more detailed regional map **B** showing the location of Sulsseewli circled in red (Map A is drawn from data from Natural Earth, map B is drawn data from Federal Office of

Topography swisstopo). **C** Photo of the lake taken upslope from the northwestern side (I. G. Alsos). **D** Schematic of elevational vegetation belts in Switzerland (modified and redrawn from Theurillat and Guisan 2001 as adapted by Tinner and Ammann 2005[10]).

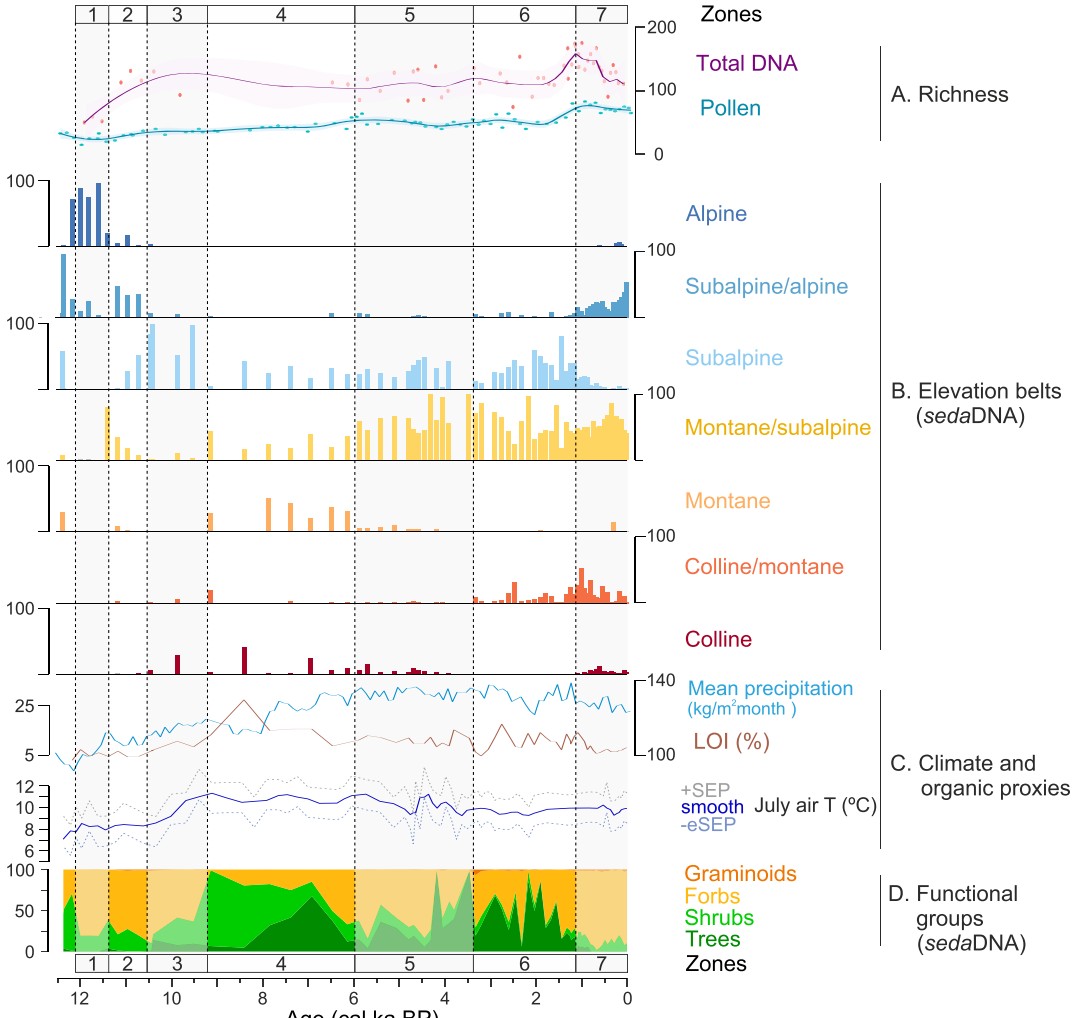

**Fig. 2 | Development of the vegetation surrounding Lake Sulsseewli during the Holocene.** Total plant richness based on strict quality filtering of *seda*DNA, as well as total plant richness based on pollen (**A**) smoothed with a loess-smoother (span = 0.1); proportion of *seda*DNA plant taxa characteristic of vegetation belts ordered from high to low elevation (**B**); mean precipitation and organic content of the sediments (as loss on Ignition (LOI)), chironomid reconstructed temperature (dotted lines indicate unsmoothed values of the sample-specific estimated standard errors of prediction, +SEP, −eSEP; the blue line the record smoothed with a 3 sample running average) (**C**); relative proportions of plant *seda*DNA representing different functional plant groups (**D**). Vegetation zones 1–7 based on constrained cluster analysis (CONISS) of *seda*DNA are indicated by vertical lines.

show that subalpine vegetation in this region has been shaped by climatic change during the Holocene but also by millennia of low-intensity agropastoral land use practices and argue that, for mountain regions with a long history of human intervention, such as the European Alps, moderate levels of human management may be needed to preserve current high plant diversity of subalpine and alpine ecosystems in the face of ongoing climate warming.

## Results

### Sediment core lithology and chronology

The lake recorded a long sedimentary sequence encompassing the entire Holocene and the youngest parts of the Late-glacial period. The core has three sedimentary units: unit I is composed of clay (706–530 cm), unit II of organic silt with clay (530–470 cm) and unit III of brown gyttja with silt laminations (470–0 cm) (Supplementary Fig. 1). The age-depth model suggests that the lowest sections of the core may be as old as 13.5 ka (Supplementary Fig. 2 and Supplementary Data 5, 6). However, the age depth model has large uncertainty below ~11 ka. This section should therefore be interpreted with caution and we focussed the analyses and interpretation on the chronologically well-constrained interval <11 ka.

### Holocene temperature and precipitation development

The chironomids from Sulsseewli indicate several major changes in temperatures (Supplementary Fig. 3 and Supplementary Data 11). To provide a local temperature reconstruction for Sulsseewli the compositions of chironomid communities were translated to estimates of past summer temperature using a chironomid - July air temperature transfer function[23] that has previously been developed based on the modern distribution of chironomids in small lakes in Switzerland and Norway and that has been extensively used and tested in the European Alps[24]. Reconstructed mean July temperatures range between 7 and 11 °C (Fig. 2C and Supplementary Data 12). The earliest Holocene presents a phase of relatively cool inferred temperatures (mean $T$ = 7.9 °C, before 11 ka), which was followed by an extended phase of relatively warm temperatures in the mid-Holocene (mean $T$ = 10.8 °C, 9.2–5.5 ka) and again cooler temperatures in the late Holocene (mean $T$ = 9.7 °C from 4 ka). This temperature development is in agreement with other climate records from the Alps[24,25]. Modelled precipitation ($P$) ranges between 103 and 131 kg/m² month (Fig. 2C). The early Holocene starts with the lowest values ($P$ mean = 105 kg/m² month; before 11 ka), followed

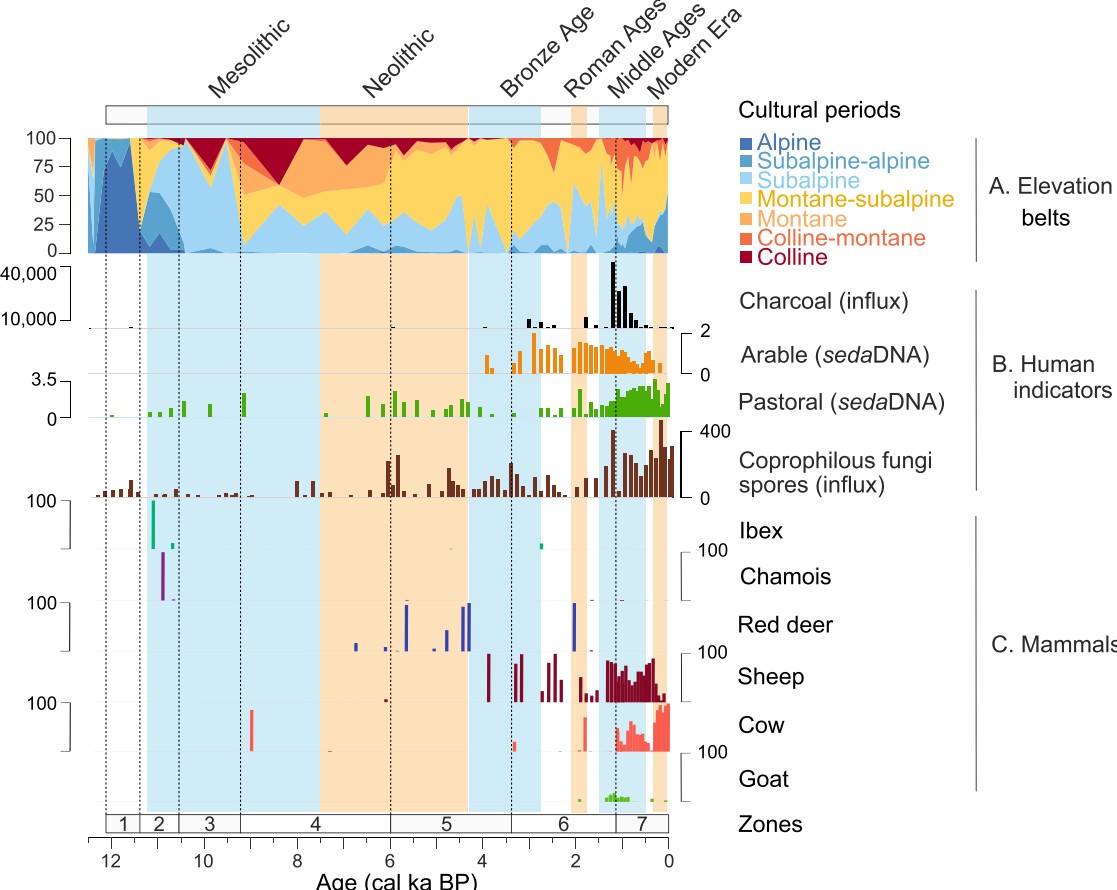

**Fig. 3 | Expanding human impact and mammalian *seda*DNA through the Holocene at Lake Sulsseewli.** For comparison cultural periods are indicated at the top of the diagram together with the relative abundance of *seda*DNA plant taxa characteristic of elevation vegetation belts ordered from high to low elevation (**A**). Human indicators (**B**): microcharcoal, arable plant *seda*DNA, coprophilous fungi spores and pastoral plant *seda*DNA, **C** mammal *seda*DNA: Ibex (*Capra ibex*), Chamois (*Rupicapra rupicapra*), Red deer (*Cervus elaphus*), Sheep (*Ovis aries*), Cow (*Bos*

*taurus*), Goat (*Capra hircus*). Vegetation zones 1–7 based on constrained cluster analysis (CONISS) of *seda*DNA are indicated by vertical lines. Vertical coloured bars correspond to the Mesolithic, Neolithic, Bronze Age, Middle Ages, Roman Ages, and Modern times. Note that the sporadic detection of Cow at -12.7 ka is not plotted due to being beyond the confident age-depth model. Additional detections, including cow at 7.4 ka, are not visible due to low read counts but are visible in Supplementary Fig. 9.

by a notable rise in mid-Holocene (*P* mean = 118 kg/m² month 10.7–6.4 ka) and a plateau of stable values in the late Holocene (*P* mean = 129 kg/m² month from 6.4 ka).

## Improved identification of plants and ecological indicators from *seda*DNA

Using *seda*DNA, we demonstrate a higher taxonomic resolution than pollen analyses (Supplementary Data 2), which allowed increased detection of indicative species and a refined interpretation of vegetation dynamics. We obtained sequences from 366 unique plant taxa from 74 sediment samples of Sulsseewli with the relaxed quality filter, and 56 samples with the strict quality (see "Methods", Supplementary Data 1, Supplementary Fig. 4 and Supplementary Table 7). From the four reference databases used to identify plant taxa in our *seda*DNA data set (PhyloAlps, PhyloNorway, ArctBorBryo, and EMBL), PhyloAlps provided the highest number of sequences assigned to vascular plants overall and increased the number of taxa assigned at genus or species level by 30% compared to EMBL. After consolidating the identifications from the four reference databases, PhyloAlps identified 87% of the total sequences, followed by EMBL which assigned 12% (Supplementary Tables 3–5). A total of 90 of our 366 identified plant taxa were informative of land use or vegetation belts: 86 were identified as indicators of specific elevational vegetation belts ("Methods", Supplementary Table 2) (Fig. 2B), one indicated arable land (*Myosotis*

*arvensis*), eight taxa were used for pastoral inference (Supplementary Table 6, Fig. 3B), and five taxa are indicators for both vegetation belts and pastoral inference. All *seda*DNA results are expressed as the relative abundance index (RAI), which integrates information from the relative proportion of reads and replicability of metabarcoding PCRs (see Material and methods), except Supplementary Figs. 4 and 9. Seven vegetation zones were obtained from the constrained cluster analysis (CONISS) of *seda*DNA.

## Pollen, plant and mammal *seda*DNA as human indicator proxies
Pollen analysis identified a total of 173 pollen types and spores (Supplementary Fig. 5). Results reveal the main vegetation changes captured by other pollen records in the region including the increase of *Pinus cembra*-type and thermophilous tree pollen (*Corylus, Ulmus, Tilia*) in the early Holocene (11–8 ka), the latter representing the upslope transport of pollen from lower elevation sites[26], and the appearance of *Abies alba* and later *Picea abies* in the mid-Holocene (8–5 ka).

We obtained 36 unique sequences from the mammalian *seda*DNA analysis (Supplementary Table 8). After removing sequences of off-target, non-mammalian, human, or pig/boar origin, we retained 14 sequences belonging to either domesticated taxa (cow: *Bos taurus*, sheep: *Ovis aries*, goat: *Capra hircus*, horse: *Equus caballus*) or wild animals (ibex: *Capra ibex*, red deer: *Cervus elaphus*,

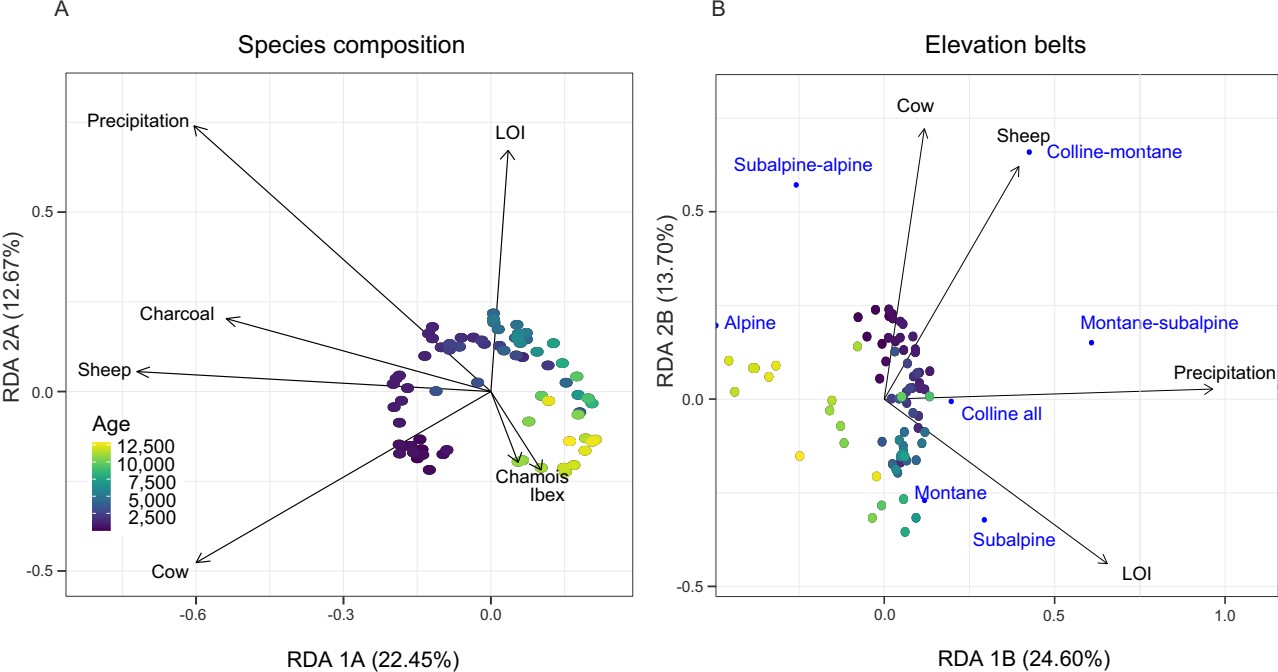

**Fig. 4 | Effects of explanatory variables on plant diversity.** Redundancy analysis (RDA) of the plant species composition (**A**) and elevational vegetation belts (**B**) from *sed*aDNA across the retained explanatory variables representing human activities (domestic mammal DNA and charcoal), climatic changes (model-inferred precipitation, and organic matter content (LOI)) and wild mammal DNA. Small blue circles represent the elevation vegetation belts (**B**).

chamois: *Rupicapra rupicapra*) (Fig. 3C; Supplementary Fig. 9; Supplementary Data 3, 9). The plant *sed*aDNA indicators reflecting arable land and pastoral activities, as well as the presence of domestic livestock *sed*aDNA, microcharcoal and coprophilous fungi spores indicate significant local human activities in the vicinity of Sulseewli during the Neolithic, Late Bronze Age, Roman Period, Middle ages and Modern Era (Fig. 3).

**Determining drivers of vegetation changes**

We conducted a redundancy analysis (RDA) of the plant *sed*aDNA data across all samples and 12 ecological variables to assess how the relative abundance index (RAI) composition of plants changed in relation to potential drivers, such as human activities (represented by coprophilous fungal spores, *sed*aDNA of the domesticated animals cow, goat, sheep and horse, and microcharcoal) and climatic changes (represented by chironomid-inferred temperatures, model-inferred precipitation, organic matter content, and the wild animals chamois, ibex, and deer) (Supplementary Fig. 12A). After removing non-informative variables, only nine were retained (Fig. 4A). Temperature was removed by the ordistep analysis but was correlated with other climatic variables retained in the analysis (precipitation; Fig. SF12). We also explored piecewise RDA, however the results did not differ from those of classical RDA (see Supplementary Fig. 13, Supplementary Data 10). The RDA of the plant compositions shows that the explanatory variables explained 42% of the total variance in species composition. The first axis (RDA 1A) explains 22.45% of total variance and is a combination of multiple independent variables related to human influence (sheep, cow, and/or charcoal). This axis also temporally separates the Early-Mid Holocene samples from the younger samples. The second axis (RDA 2A) explains 12.67% of the total variance and is related to variables reflecting climate (sediment organic matter content or precipitation), as well as *sed*aDNA of wild animals (chamois/ibex, only with relaxed filtering). We also performed RDA of the elevational vegetation belts against the ecological variables to assess how the RAI composition of temperature-sensitive plants changed in relation to potential drivers (Supplementary Fig. 12B). Only four explanatory variables were

retained as informative (cow, sheep, precipitation, and organic matter content), explaining 41.34% of the total variance (Fig. 4B). The first axis (RDA 1B) explains 24.60% of the total variance and is related with climate indicators (precipitation/organic matter content). The second axis (RDA 2B) explains 13.70% of the total variance and is related to human-activity (cow/sheep pasturing). To explore the potential predictors of plant richness, we used generalised additive models (GAMs). We used our "strict" filtered data for GAMs, which showed significant non-linear patterns between plant richness and three drivers (precipitation, sheep, goat) (Fig. 5; Supplementary Table 10). Plant richness increases along the precipitation gradient initially, then decreases, and finally remains stable (Fig. 5A). The presence of sheep and goat at intermediate RAI values (~60%, ~10%, respectively) seems to promote plant richness, whereas sheep at high values decrease richness (Fig. 5B, C). Furthermore, we used a structural equation model (SEM) with the "strict" filtered data to explore how and to what extent the predictors influence on plant richness can be modelled by this approach (Supplementary Table 11 and Fig. 6). The model fit the data reasonably well (Fisher C = 6.259, df = 18, p-value = 0.995). Results show that plant richness ($R^2 = 0.45$) can be modelled as being significantly and positively affected by precipitation and charcoal. On the other hand, SEM also models precipitation as having an indirect effect on sheep and a negative indirect effect on ibex and chamois, while charcoal has an indirect effect on sheep (Supplementary Table 11). Consistent with our RDA analysis that shows both human and climate drivers of plant community, GAM and SEM analyses show that both climate and human related predictors significantly explain plant richness, with an increasing influence on the human part. Given that both strict and relaxed RDA analyses on plant community broadly separate the Early-Mid from the Mid-Late Holocene samples at ~6 ka, we split our description of the proxy results into the two periods highlighted by RDA 1A.

**Climate-associated vegetation changes during the Early to Mid-Holocene**

The vegetation composition and the elevational belts varied along the climate in the RDA analysis (organic matter content, precipitation)

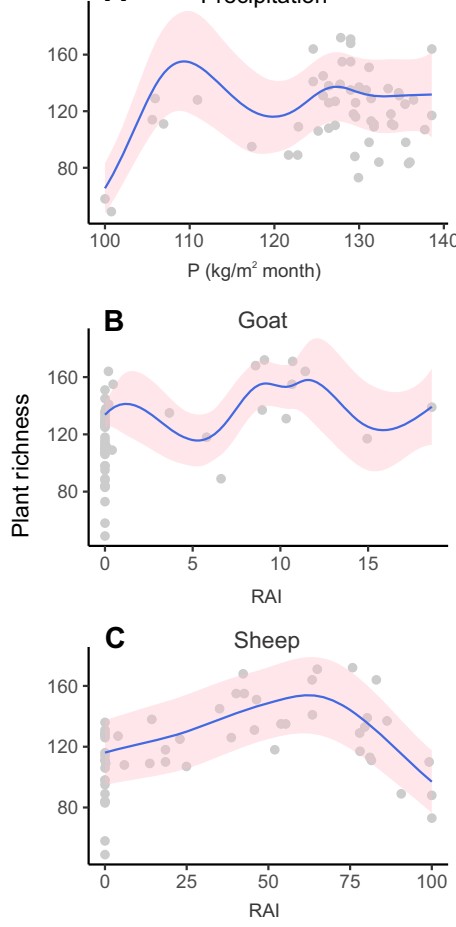

**Fig. 5 | Results of the GAM prediction.** Relationship between plant richness and three explanatory variables that were determined to be statistically significant (Supplementary Table 10): **A** Precipitation (kg/m² month), proportion of **B** goat (RAI) and **C** sheep (RAI) based on the GAM prediction. The blue lines and pink shadings represent mean GAM prediction and their ± 1.96 standard errors respectively. Points in the background are for observed data.

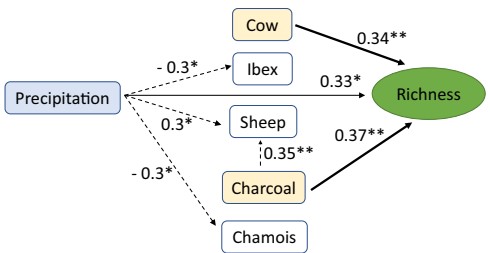

**Fig. 6 | Selected structural equation model by using the function gls on the strict quality dataset with standardised coefficients.** Arrows represent the unidirectional relationships among the responses and predictors. The thickness of the paths (arrows) is scaled to the magnitude of their significance. Dashed lines show indirect effects of precipitation and charcoal. Fisher C = 6.259, *df* = 18, *P* value = 0.995. Significance codes: "**" 0.01, "*" 0.05.

until 6 ka (Fig. 4A, B; RDA 2 = 12.67%; RDA 1 = 24.60%). Dating the oldest analysed sediments of the Sulsseewli record is challenging with the available chronological and sedimentological data (Supplementary Fig. 2), but they likely originate from around the Younger Dryas interval. Low July temperatures and precipitation are inferred for these lowest sediment sections (Fig. 2C), which is consistent with the

transition from the Younger Dryas into the Early Holocene[27]. Based on plant *sed*aDNA, this period was associated with a high abundance of cold-adapted Alpine forbs such as *Achillea atrata*, *Crepis rhaetica* and *Silene acaulis*, with some *Dryas octopetala* dwarf shrubs (Fig. 2D and Fig. SF4). This period had the lowest values of organic matter content, suggesting limited vegetation cover and organic production for this interval (Fig. 2C, Zone 1, 12–11.35 ka, T range = 7–9 °C). An increase in Subalpine/Alpine plant taxa (*Athamanta cretensis*, *Chaerophyllum villarsii* and *Carex frigida*) occurs with a plateau of the inferred temperature of ~8.5 °C at 11.35–10.55 ka (Fig. 2B, Zone 2). This resulted in the most significant forb introduction in the catchment (~50 taxa) and the highest diversity of *sed*aDNA plant species for the Early Holocene (~90 taxa per sample) (Supplementary Fig. 8). The number of forbs in our *sed*aDNA results align with the high proportions of Poaceae and *Artemisia* pollen recorded during the earliest Holocene (Supplementary Fig. 5, pollen Zone SUL-1). However, the lower taxonomic resolution of forb, grass and sedge pollen (96 taxa) does not reveal the exceptionally high diversity of the herb community that is recovered by *sed*aDNA (232 taxa) or the full breadth of cold-adapted species, and shows lower richness than indicated by *sed*aDNA throughout the Early to Mid Holocene (Fig. 2A, Supplementary Fig. 8, and Supplementary Data 2). During this earliest Holocene interval, we observe only sporadic mammalian *sed*aDNA sequences of two wildlife taxa, ibex and chamois, suggesting a low mammalian biomass and a natural landscape at this time (Fig. 3C and Supplementary Fig. 9).

Temperatures then increased and reached a second plateau by ~9.5 ka with values that were as much as ca. 3 °C higher than in the Younger Dryas interval. This represents the most dramatic temperature increase in our record and, based on plant *sed*aDNA, co-occurs with the upward migration of Subalpine taxa like *Crepis bocconei* whereas Subalpine-Alpine taxa almost disappear, and richness remains stable in the vicinity of the lake (Fig. 2A, B, Zone 3, 10.55–9.2 ka). The warmer and relatively stable temperatures and rise of precipitation during the Holocene Climatic Optimum (9–5.5 ka, ~11 °C) coincides with the upward expansion of Montane-Subalpine taxa (*Abies alba*, *Chaerophyllum aureum*, *Lonicera alpigena*), along with a dominance of woody taxa (Supplementary Figs. 4, 5 and Fig. 2B, D, Zone 4, 9.2 to 6 ka). The rise of shrubs (Maleae, Rosoideae), which began at ~10.5 ka co-occurs with the highest values of sedimentary organic matter content (Fig. 2A, C, D). As shrubs expanded, the RAI of herbs decreased and diversity dropped to only a third of the taxa that dominated in the earliest Holocene (~65 taxa) (Fig. 2A and Supplementary Fig. 7). We recovered a single occurrence of domestic cow (*Bos taurus*) at ~9 ka, which we attribute to stochastic contamination (see "Methods") as it only occurs in a single sample and there is no archaeological evidence to support this occurrence[28]. A rise in coprophilous fungi spores between 7.9 and 7.15 ka, without a notable change in the sediment accumulation rate (Supplementary Fig. 2), suggests a greater abundance of mammalian herbivores and therefore higher grazing pressure in the vicinity of Sulsseewli (Fig. 3B, C). The lack of arable and pastoral indicators and domesticated livestock *sed*aDNA suggest that this may have resulted from wild animals. The decrease in trees and shrubs in the plant *sed*aDNA record beginning at 7 ka may have facilitated the introduction of new herb species (Fig. 2A, D and Supplementary Fig. 8). During this interval, red deer (*Cervus elaphus*) are first detected in the record.

## Vegetation change with increasing human activity in the Mid-Late Holocene

After 6 ka, vegetation change is increasingly associated with indicators of human impact (Fig. 4A). The first major evidence of human disturbance through grazing appears around 6 ka (Fig. 3), as inferred from the common rise of several indicators of human activity (micro-charcoal, coprophilous fungi spores, plant *sed*aDNA indicating pastoral activities) (Fig. 3B, C). The first cereal pollen appeared around this

time (Supplementary Fig. 6), indicating agricultural practices in the vicinity of the Sulsseewli catchment, presumably at lower elevations in the main Lauterbrunnen valley (Fig. 1A). An increase in grazing pressure is evidenced by a rise of coprophilous fungi spores, pastoral plant *sed*aDNA, and microcharcoal during the late Neolithic (Fig. 3B). This human impact signal continues in the interval between the late Neolithic and earliest Bronze Age, 5.7–4.2 ka and alternated with periods of less human influence and multiple detections of red deer as well as a rise of mesophilous, Montane-Subalpine plant taxa such as *Saxifraga rotundifolia*, *Vicia sylvatica* (indicative of forests with open canopies), and *Trollius europaeus* (wet meadows) by 5 ka (Fig. 2B, Zone 5, Supplementary Fig. 5).

From the start of the Bronze Age (ca. 4.2 ka) onwards, the continuous presence of diverse indicators of human activity suggests that Sulsseewli has been exploited as a grazing area for domesticated livestock. This interpretation is supported by an increase in pollen and *sed*aDNA of species favoured by human disturbances such as *Alnus alnobetula* and *Plantago lanceolata* (Supplementary Figs. 4 and 5). A short period of agricultural activity at 3.9–3.8 ka during the Early Bronze Age is marked by the first appearances of the arable plant *sed*aDNA indicator and cereal pollen including subtypes Cerealia, *Hordeum* and *Triticum* (Fig. 3B and Supplementary Fig. 6). This period also includes the first occurrence of sheep *sed*aDNA, which is suggestive of pastoralism. The Late Bronze Age (3.35–2.8 ka) is characterised by intensifying human impact. The rise of indicators of human impact suggests that humans burned forests and/or grasslands (microcharcoal), cultivated crops (arable plant *sed*aDNA and pollen of Cerealia and *Hordeum*, Supplementary Fig. 6), and pastured sheep (*sed*aDNA, coprophilous fungi spores) in the Montane-Subalpine plant communities nearby Sulsseewli, although we note the potential disappearance of sheep in the latest Bronze Age (Fig. 3B, C). After the Bronze Age, high microcharcoal concentrations together with high RAI values of sheep and arable *sed*aDNA point to recurrent burning to maintain nearby pastures and crops (Fig. 3B). At this time, the proportion of Colline-Montane plant taxa increased (Fig. 2B). However, a drop in all human indicators and the reappearance of red deer, a wildlife animal, around 2.2–2 ka and preceding the Roman period suggests the temporary abandonment of agropastoral activities in the zone. This coincides with Subalpine plant taxa and conifer reforestation (*Pinus cembra*, *Picea*, Supplementary Figs. 4 and 5), leading to another decrease in meadows (Fig. 2D). By the Late Roman period (ca. 1.6 ka), a marked decrease of trees reflected by plant *sed*aDNA functional groups and increased microcharcoal concentrations suggest a high fire incidence to clear the forest. The high proportions of pastoral plant *sed*aDNA and co-occurrence of goat, sheep and cow *sed*aDNA also support that forest was cleared for the pasturing of diverse livestock (Fig. 3B, C). After the Roman period, human activity continued with the presence of arable *sed*aDNA, pollen from cereals (Cerealia, *Hordeum*, *Triticum* and *Secale*), *sed*aDNA of Colline cultivated trees (*Juglans regia*, *Castanea sativa*), and a continued decrease in tree pollen and *sed*aDNA (Supplementary Figs. 4–6). However, *sed*aDNA of domestic livestock taxa was not detected during this brief interval prior to the Middle Ages.

The Middle Ages (1.4–0.5 ka) saw a massive increase of microcharcoal and a near loss of tree pollen and *sed*aDNA (Figs. 2D and 3B), suggesting high fire activity to clear the local forest[2]. The rise of coprophilous fungal spores, plant *sed*aDNA pastoral indicators, as well as sheep, goat, cow, and horse *sed*aDNA also points to large-scale grazing (Fig. SF4). Total plant diversity (~160 taxa) and the abundances of both Subalpine-Alpine and lowland plant species rise at this point (Fig. 2A–D, Zone 6, Supplementary Figs. 8 and 5). A contribution from higher elevation taxa could have been favoured by a reduction in competition from shrub and tree species in open meadows, together with greater erosion due to deforestation that might have favoured transport of high-elevation plant *sed*aDNA to the lake. This scenario is

supported by the very high sedimentation rates in the uppermost sediment layers that are consistent with high erosional input (Supplementary Fig. 2). The rise of grazing activities coincides with the proliferation of lowland taxa and a rise of the total floral diversity (Fig. 2A, B, D). The greater influence from lower elevation vegetation would have led to a contraction of space and increased competition for Alpine species (Fig. 2B). Finally, the recent increase in plant *sed*aDNA of Subalpine-Alpine taxa during Medieval and Modern times (1.2–0 ka) coincides with some Colline-Montane taxa (Fig. 2B, Zone 7). Increased human activity and grazing may have reduced the standing biomass of the more competitive Subalpine taxa (grasses, dwarf shrubs), allowing heliophilous, shorter Subalpine-Alpine taxa to occur at a lower elevation, resulting in an overall increase in species richness (Fig. 2B, D). A rise of Alpine and Subalpine/Alpine taxa together with a decrease of July air temperature is observed by 1700 CE (Fig. 2B, C), coinciding with periods of glacier advances in the central Alps during the Little Ice Age[29,30]. The highest values of coprophilous fungal spores, together with plant *sed*aDNA pastoral indicators and cow-dominated *sed*aDNA suggest a return to high mountain grazing during the 18th Century CE (Figs. 2 and 3).

## Discussion

The diverse multiproxy and high resolution approach of this study makes it the most detailed palaeoecological reconstruction of Holocene Alpine vegetation to date. With 366 identified plant taxa, Sulsseewli yields the richest single record studied with *sed*aDNA using an exact match. This is 2–3 times as many taxa as found in any previous study of ancient plant *sed*aDNA[6,7,31,32], and highlights the advantage of using a region-wide comprehensive, rather than global or incomplete regional reference databases[8,33,34]. Our *sed*aDNA results uncover a hidden diversity of taxa that are normally underrepresented by pollen analysis[35–37], including many insect-pollinated plants, and highlights the relevance of forbs in Subalpine communities[38]. The pollen of Sulsseewli (166 vascular plant taxa) present a similar diversity and main vegetation changes as nearby lakes Sägistalsee (210 taxa)[39], Bachalpsee (179 taxa)[40], Iffigsee (143 taxa)[2] and Lac de Bretaye (137 taxa)[41]. These results suggest that the high values of richness found in Sulsseewli may be representative of the area and not only related to good DNA preservation and recovery, thereby indicating that the study region has been a plant diversity hotspot over the entire Holocene.

The high number and resolution of detected taxa allowed us to identify those that are informative about elevational vegetation belts and therefore reconstruct how Alpine, Subalpine and Montane plant communities changed throughout the Holocene at Sulsseewli (Fig. 4B), which is coherent with the regional trends of pollen and macrofossil studies[26,40,42]. Pollen studies reconstructed the shifts of the Montane-Subalpine ecotone by grouping characteristic pollen taxa, with the limitation that pollen taxa can often include several species and autecologies[2,43]. Some *sed*aDNA studies used single genera such as *Plantago* to infer human activities[8,33]. However, this genus has ~16 species that are found in the Alps, of which only four are favoured by grazing (*P. alpina*, *P. lanceolata*, *P. media*, *P. major*)[22,44]. On the other hand, Liu et al.[6] used single species such as *Sanguisorba officinalis* to infer human activities in the Tibetan Plateau. However, this species can also be found in undisturbed sites such as wetlands or meadows. Our *sed*aDNA approach provides a more robust inference since its ecological interpretations are based on groups of well-characterised taxa. We have also overcome the taxonomic limitation of some species sharing the same marker sequence[45] by retaining the sequences as indicators only if their autecological category is shared among all species assigned to that sequence (see "Methods"). As more high-resolution data like those generated here become available, our understanding of the spatial dynamics of specific taxa and rates of extinction will increase and it may be possible to detect new refugial areas. The information obtained from these stricter indicator

sequences opens up a range of possibilities to robustly reconstruct unexplored parameters in palaeoecology. For example, Ellenberg or Landolt indicator values for the central-European flora can be used to estimate site conditions from species composition when other measures of environmental variables are not available (e.g., nutrient availability, nitrogen, light availability)[46].

The rise of DNA richness in Sulsseewli during the Early Holocene contrasts with the minor increase observed in palynological richness in both Sulsseewli and the wider Alps[47] (Fig. 2A). In this study, the higher representation of forbs by sedaDNA fills a knowledge gap about herbaceous taxa that could not have been resolved by pollen or might have been masked by high pollen-producing taxa (Supplementary Fig. 7). Our sedaDNA results paint a more detailed picture of plant diversity than the pollen results during the Early Holocene. Richness significantly responds to precipitation at the beginning of the Holocene (Fig. 5A and Supplementary Table 10). Thus, the more detailed plant record detected with sedaDNA than pollen may allow a deeper understanding of the relationship between plant diversity and climate change.

Our multiproxy palaeoecological approach also reveals that, from the Neolithic onwards, there is an increased co-occurrence of taxa dwelling at different elevation belts, possibly promoted by increased erosion following deforestation (Fig. 2B, D). This is associated with a rise in diversity in the vicinity of Sulsseewli (Figs. 2A and 5), a change also supported by a clear increase in regional pollen diversity[48,49]. Considering the strong evidence of intensified grazing, it seems likely that grazing activities during the Neolithic modified the structure and composition of the vegetation by patchy disturbance, dung deposition and grazing preference. This is in line with social and demographic changes that occurred in the Late Neolithic population[50] associated with a rise in transhumance activities in the region and a downward shift in treeline[40,51–54]. The input of nutrients and seeds by livestock when moving across valleys and high mountains during seasonal transhumance[55] could have introduced new species that might have been able to grow in the new microhabitats created around the lake (e.g., nutrient-rich depressions, open areas suitable for pioneer taxa)[49,56,57]. Indeed, part of the pastoral plant sedaDNA indicators are species dispersed by animals[22]. The shift towards combined arable and pastoral farming during the Bronze Age (3.4 ka) prompted a further disruption of the landscape, where sheep pastoralism, fires and deforestation favoured the presence of Colline-Montane taxa. This trend agrees with a rise of pastoralism in nearby lakes: Lac de Bretaye (68 km south-west of Sulsseewli)[41], Sägitalsee and Bachalpsee (10 and 14 km north-east respectively)[40]. Sulsseewli offers the earliest robust evidence of domesticated livestock DNA in the Northern Swiss Alps, which occurs at least 1000 years earlier than in the western Alps (Lake Anterne, Iron Age, 2.5 ka[58]) although plant grazing indicators suggest that the first significant human impact might also have occurred as early as 3.5 ka in that region of the Alps[33]. The shift towards mainly cow-dominated livestock farming and deforestation by 700 years ago coincides with the establishment of an Augustinian monastery in the Interlaken valley (20 km from Sulsseewli), which might have facilitated the intensification of pastoral practices in the region[8,40] as part of the cheesemaking economy[59,60].

Our study showcases how late Holocene changes in plant composition and richness around Sulsseewli were associated with increasing human activities, leading to a strongly structured landscape. Based on the intermediate disturbance hypothesis[61,62], for vegetation that is not limited by moisture availability, such as in the European Alps, moderate levels of disturbance are expected to lead to the highest diversity of habitats and plant species. Indeed, we found a hump-shaped relationship between sedaDNA of sheep and plant richness. Further, agropastoral activities during the last millennium may have maximised spatial heterogeneity at Sulsseewli and, consequently, increased plant species diversity. This is corroborated by contemporary studies that relate land abandonment during the past

decades to a significant reduction in plant diversity[63] as well as by observations of peak plant richness with moderate levels of grazing[62]. Therefore, our long term reconstruction of plant diversity based on sedaDNA and other palaeoecological indicators confirms earlier studies that suggested that maintaining high plant diversity in mountain ranges such as the Alps, whose vegetation has been shaped by a long history of human land use, requires conserving moderate land use practices[13,64,65]. This contrasts with Liu et al.[6,66], who suggest that grazing effects might be too weak to compensate for climate warming impacts[6,66]. This discrepancy might be explained by lower human impact at their study site, which is located at a much higher elevation (4200 m a.s.l). Similarly, the situation may differ for mountain vegetation stressed by limited moisture availability, where even relatively low levels of disturbance may lead to loss of plant diversity[67]. However, our findings support palaeoecological and contemporary studies that registered the potential of grazing to mitigate the elevational advance of climate-driven shifts in plant communities through increasing total stress[49,68]. Furthermore, recent studies suggest that shifts from intensive to extensive agricultural land use can favour native species while controlling the oversimplification of the flora due to plant invasion[66]. Our interpretations should be supported, and potentially refined, by additional studies that assess the effects of different degrees of grazing intensity on alpine vegetation under variable biogeographic, temperature, and particularly moisture regimes. Similarly, the effects of extensive agricultural land use may differ, and potentially be more negative, on other elements of landscape diversity, such as the diversity of mammals or other organisms in mountain regions[69,70].

A rise in plant diversity in alpine systems similar to Sulsseewli might be a sign that land-use allows the coexistence of species adapted to different habitats. Many studies point out that regional plant richness may increase with warmer temperatures[10], prompting an acceleration of changes in mountain summits during the past century due to the upward movement of lower vegetation belts[3]. In the view of our results, we question the use of plant total diversity as a measure of ecosystem health in alpine systems and consider the introduction of other diversity metrics. In contrast with Liu et al.[6], we propose metrics that are based on particular elevational vegetation belts, as demonstrated here, instead of using the richness within the most common alpine plant families since they can contain taxa belonging to different elevational habitats and communities. Future studies should also consider the impact of grazing on the total biodiversity network including herbivores, pollinators, and predators[69].

In conclusion, the local and regional plant richness of the Subalpine-Alpine landscape in the European Alps is strongly related to millennia of agropastoral activities. Currently, high elevation farms and pastures are increasingly abandoned. However, maintaining such activities may be key for preserving the high plant diversity of alpine landscapes that is now threatened by climate change[17,18,48,63,66]. Intermediate levels of disturbance associated with these extensive land use practices could ensure the continuity of the ecological niches for slow-growing alpine species, and limit the competition that they are facing by the upward shift of more competitive species from lower elevation[3,4,10,71].

## Methods
### Study site
Sulsseewli is a small lake (2 ha) in the Bernese Alps (northern Swiss Alps), located in the subalpine vegetation zone below the present treeline (Fig. 1, 46.617639° N, 7.864028° E; 1921 m a.s.l.). The catchment lies on calcareous bedrock, surrounded by bare rock and scree slopes, together with meadows and a scattered spruce (Picea abies) forest on the eastern side. Sulsseewli has two water inlets and a subaquatic outlet, with a maximum water depth of ca 7.5 m. This site has a subalpine or boreal climate with cool summers (10 °C July) and cold winters (−5 °C January)[21].

Summer farming at elevations above ca. 1800 m a.s.l. established during prehistoric times[54]. Historical records show that alpine farming was related to the production of dairy goods and only exceptionally to grow crops such as cereals or vegetables, because the families of the summer herders were living in the valley floor, where the fields were located[72]. After 1500 AD, agricultural self-sufficiency was replaced by an overall specialisation on dairy products that were exported to the markets of the Alpine forelands, so that cereal production in the Bernese Alps was strongly reduced or abandoned[73]. This system lasted until the 1950s, when a globalised economy led to the strong reduction of farming activities in the Alps, releasing afforestation processes[74].

## Core sampling, lithology, and chronology

Four parallel cores (SUL A, B, C, D) were retrieved in August 2018 with a modified Streif-Livingstone piston corer[75]. Sediment cores were sealed in the field to avoid contamination. Sediments were then extruded in half-pipes and immediately sealed to minimise the risk of modern DNA contamination at the Institute of Plant Sciences & Oeschger Centre for Climate Change Research (University of Bern, Switzerland). Cores A and B were stored at the University of Bern, while cores C and D were stored at the Arctic University Museum of Norway in Tromsø (TMU, Norway). All cores were stored at 4 °C. At both facilities, core sections were opened by longitudinal splitting. One half was used for proxy subsampling, and the other half was used for photography. The cores were correlated based on lithostratigraphic features visible in overlapping core segments (blue lines, Supplementary Fig. 1). A composite master core of 716 cm without hiatuses was then constructed based on the cores SUL A and B (red lines in Supplementary Fig. 1 indicating tie points in the master sequence). Core C was correlated to the master sequence based on lithological marker layers and analyses of this core are presented on the depth scale of the master core based on this correlation, whereas SUL D was not used. Correlation points between the different cores as well as the master sequence are shown in detail in Supplementary Fig. 1.

The organic matter content of the sediments was measured by mass loss-on-ignition (LOI) following the Lamb method[76]. Samples were taken at the same intervals as sedaDNA, dried overnight at 105 °C, and burned at 550 °C for 2 h to oxidise the organic matter. Total LOI was calculated as the percentage loss of dry weight after ignition.

Twenty-three plant macrofossil remains were sampled and radiocarbon-dated using accelerator mass spectrometry (AMS) at either the Laboratory for the Analysis of Radiocarbon at the University of Bern ('BE' accessions) or the Poznan Radiocarbon Laboratory ('Poz' accessions, Poland) (Supplementary Table 1). AMS radiocarbon dates were calibrated using the terrestrial IntCal20 curve[77] within an age-depth model that was constructed using the Bayesian framework calibration software 'Bacon' v.2.3.4[78] in R v3.4.2 (R Core Team 2017).

## Chironomid-based temperature reconstruction

For chironomid analysis, 75 sediment subsamples of 0.125–9 cm³ were sieved (100 μm) from the correlated composite section of cores SUL A and B without chemical pretreatment following Brooks et al.[79]. Chironomid head capsules were recovered from the sieve residue with a fine forceps under a stereomicroscope at 30–50× magnification and mounted on slides in Euparal mounting medium. Identification was performed by examining the mounted specimens under a compound microscope (100–400× magnification) and using keys for Chironomidae larvae[79,80].

A 274-sample modern chironomid-temperature calibration dataset is available from Switzerland and Norway[23], describing the distribution of chironomid taxa in small lakes relative to observed mean July air temperatures. As described in Heiri et al.[23], these calibration data were used to develop a Weighted-Averaging-Partial Least Squares (WA-PLS) regression model that predicts mean July air temperatures based on the composition of fossil chironomid assemblages and that is here used to reconstruct past July air temperature development from the Sulsseewli chironomid record[23]. The model, based on two WA-PLS components and square root transformed chironomid assemblage data, predicts mean July air temperatures with a root mean square error of prediction of 1.40 °C and an $r^2$ of 0.87 in the modern environment (estimated based on cross-validation (bootstrapping) with 9999 bootstrapping cycles). Before application to the downcore record several samples with low counts were amalgamated with adjacent samples to result in a higher count size for reconstruction. The final record consisted of 73 samples with a minimum count sum of 40 chironomids, except for a single sample at 389.5 cm sediment depth (36 chironomids). Sample-specific estimated standard errors of prediction were calculated using 9999 bootstrapping cycles. The transfer function and reconstruction were developed with the programe C2[81]. The temperature values corresponding to the sedaDNA ages were obtained using nearest neighbour interpolation.

## Precipitation time series

Local mean monthly precipitation for the last 12 ka were estimated from the monthly precipitation of CHELSA-TraCE21k model[21]. CHELSA-TraCE21k provides 1 km resolution climate data for precipitation in 100-year time intervals for the last 21,000 years. Paleo orography at high spatial resolution and at each timestep is created by combining high resolution topography and information on the present, historical glacial cover with the interpolation of a dynamic ice sheet model (ICE6G), and a coupling to mean annual temperatures from CCSM3-TraCE21k. Based on the reconstructed paleo orography, mean monthly precipitation values were downscaled using the CHELSA V1.2 algorithm to a 1 km resolution. The time series was finally extracted for the geographic coordinates of the Sulsseewli lake and precipitation values for sedaDNA ages were interpolated with nearest neighbour interpolation.

## Pollen, spores, and microcharcoal

A total of 75 samples of 1 cm³ were sampled from the composite section of cores SUL A and B for pollen, spores, and microscopic charcoal analyses. Pollen samples were processed following standard palynological approaches[82] with KOH, HCl, HF, acetolysis, and mounting in glycerine. Lycopodium tablets were added before the chemical treatment as a control and to estimate microfossil concentrations[83]. Pollen and spores were identified at 400× magnification according to Moore et al.[82] and Reille[84] and the reference collection at the Institute of Plant Sciences (University of Bern). A minimum of 500 terrestrial pollen grains per sample was counted. Up to 166 pollen types of terrestrial and aquatic vascular plants were identified. Non-pollen palynomorphs (NPPs) including coprophilous fungal spores were identified according to van Geel et al.[85] at 400x magnification. Diagrams were plotted with the programme Tilia. Statistically significant pollen zones were determined with partitioning using optimal sum of squares and the broken stick method[86]. To study the regional fire activity, microscopic charcoal was analysed[87,88]. Particles between 10 and 500 μm were counted on the pollen slides following Tinner et al.[89]. Coprophilous fungi spores were analysed to infer grazing pressure (Cercophora, Delitschia, Podospora, Sordaria, Sporormiella and Trichodelitschia). The palynological results are presented as percentages of the terrestrial pollen sum excluding aquatic plants, whereas charcoal and fungal spores were presented as influx (particles cm⁻² year⁻¹, Supplementary Fig. 5).

## Sedimentary ancient DNA data generation

Core SUL C was subsampled at 10 cm resolution in the ancient DNA lab at TMU. DNA was extracted from 80 sediment samples and 8 extraction/ sampling negative controls using a a modified DNeasy PowerSoil kit (Qiagen, Germany) protocol[7] in the ancient DNA laboratory at TMU. DNA extracts and negative extraction/sampling controls, along with 8

PCR controls, were amplified using uniquely dual-tagged universal primer sets (Supplementary Data 7) that amplify either the *trn*L P6 loop region of the chloroplast genome, a locus that has proven most successful for studies of plant *sed*aDNA[90,91], for plants (gh primers; ref. [92]) or a section of the mammalian mitochondrial 16S locus (MamP007 primers; ref. [8]). Metabarcoding PCR reaction and cycling conditions followed Voldstad et al.[93]. Reactions consisted of 40 µL final volumes containing 1X Gold buffer, 1.6 U of AmpliTaqGold DNA Polymerase (Life Technologies), 2.5 mM $MgCl_2$, 0.2 mM dNTPs, 0.2 µM of each primer, 160 ng/µL of Bovine Serum Albumin, and 4 µL of DNA extract. The following exceptions were applied for the mammalian mitochondrial 16S PCRs: (1) forward and reverse primer concentration was reduced to 0.1 µM each, and (2) forward and reverse blocking primers were added at 1 µM each. We used a forward blocking primer modified from Giguet-Covex et al.[8] with a sequence of GGAGCTTTAATTTATTAATGCA AACAGTAGG-C3 and a new reverse blocking primer with a sequence of CCCAACCGAAATTTTTAATGCAGGTTTGGTGA-C3. Cycling conditions consisted of enzyme activation at 95 °C for 10 min, followed by 45 cycles of denaturation at 95 °C for 30 s, annealing at 50 °C for 30 s, and elongation at 72 °C for 1 min, plus a final elongation step at 72 °C for 7 min. Eight PCR replicates were carried out for each sample or control for *trn*L, whereas four replicates were performed for 16S. The PCR replicates were pooled and cleaned[93], and the pools were converted into 4 DNA libraries using a modified TruSeq PCR-free library kit (Illumina) and unique dual indexing[7]. The library was quantified by qPCR using the Library Quantification Kit for Illumina sequencing platforms (KAPA Biosystems, Boston, USA), using a Prism 7500 Real-Time PCR System (Life Technologies, The Norwegian College of Fishery Science, UiT). Each library was sequenced on ~10% of a flow cell on an Illumina NextSeq 500 platform (2×150 bp, mid-output mode) at the Genomics Support Centre Tromsø (UiT).

## Database construction and sedimentary ancient DNA data analysis

The OBITools software package[94] was used for the bioinformatics pipeline, following the protocol and criteria defined by Rijal et al.[7]. Briefly, paired-end reads were aligned using SeqPrep (https://github.com/jstjohn/SeqPrep/releases, v1.2). Merged reads were demultiplexed according to the 8 bp unique primer tags (Supplementary Data 7) and identical sequences were collapsed. Singleton sequences and those shorter than 10 bp were removed and putative artifactual sequences were identified and removed from the dataset[32].

Previous metabarcoding studies of the Alps that targeted the *trn*L P6 loop have relied on global databases with a sparse representation of the Alpine flora resulting in limited, and potentially inaccurate, identifications[8,33,34]. Here we generated a local DNA reference library. We used the PhyloAlps genome skim database that comprises 4604 specimens of 4437 taxa collected in the Alps (*n* = 4280) and the Carpathians (*n* = 324) (https://data.phyloalps.org/browse/). A full description of this database including the standard barcodes ITS2, *mat*K and *rbc*L are available in Alsos et al.[91]. Here, we compiled a P6 loop database by running the ecoPCR software[95] on their corresponding raw databases with the gh primers (Supplementary Data 4). The same approach was also used for the PhyloNorway genome skim database[32,91], and the global EMBL rl143 database.

The assignment of the reads was then applied to the four different reference libraries: (1) PhyloAlps; (2) ArctBorBryo (regional arctic/boreal reference library compiled from Willerslev et al.[31], Sønstebø et al.[45] and Soininen et al.[96]); (3) PhyloNorway[91]; and (4) the global reference library based on the EMBL rl143 database. For mammals, we generated a reference library from the EMBL rl143 database as described above except using the 16S primers. The identified sequences were filtered following the method described by Rijal et al.[7], whereby only sequences with a 100% match to a reference sequence, represented by three reads per replicate, and with a minimum of 10

total reads and three replicates across the entire data set were retained. Furthermore, artificial sequences that were the result of PCR or sequencing errors, such as homopolymer length variation, were merged with the source sequence (following Rijal et al.[7]). Additionally, common laboratory contaminants and sequences that displayed higher average read counts in negative extraction or PCR controls than lake sediment samples were removed (all taxa found in the negative controls are listed in Supplementary Data 8). However, we retained some plant taxa found in the controls, such as *Pinus* and *Picea abies*, that are ecologically plausible and were independently detected in the pollen analysis. Final taxonomic assignments for plants were based on the collapsed assignments to the four reference databases, giving priority to the local PhyloAlps identifications. The identified taxa were compared with the species in Flora Alpina[44] and Flora Indicativa[22].

We initially removed six plant *sed*aDNA samples with low metabarcoding data quality, which we define as a technical quality (MTQ) score <0.45 and/or analytical quality (MAQ) score <0.175. These MTQ/MAQ score thresholds were derived from the maximum scores of the negative controls[7] (Supplementary Table 11). We evaluated the per sample distribution of other plant metabarcoding data quality metrics[7], including the (1) total raw read count, (2) mean barcode length, (3) mean proportion of weighted PCR replicates, and (4) proportion of reads identified as terrestrial plant taxa. We examined these metrics against time and plant richness (Supplementary Figs. 10 and 11). We found non-significant correlations for both total raw read count and mean barcode length against both time and richness, suggesting that sequencing effort and DNA degradation have not adversely affected our analyses (Supplementary Table 11). The remaining metrics, including MTQ and MAQ scores, had significant correlations. Although it has been noted that these metrics are not necessarily independent of richness[7], their correlation with age suggests potentially reduced detectability of plant taxa in older samples.

Given the significant associations between sample age and four of the six measures of plant *sed*aDNA data quality, including both MTQ and MAQ scores (Supplementary Figs. 10, 11), we additionally divided our retained samples into two groups: those that passed a "relaxed" filter (as previous, MTQ ≥ 0.45, MAQ ≥ 0.175; *n* = 74 samples) and those that passed a "strict" filter (requiring both MTQ and MAQ scores ≥0.9; *n* = 56 samples). This latter filter removed both an additional 18 samples and the significant trends observed between all data quality metrics and sample age, with the exception of mean wtRep which undergoes a shift in samples beginning c. 3.0–2.0 ka (Supplementary Table 9; Supplementary Fig. 10). We emphasise that the mean wtRep trend is otherwise flat between 12.0 and 3.0 ka, which does not support the hypothesis of an age-induced decrease in plant *sed*aDNA data quality in the strict dataset. We note that both filtering strategies gave the same broad RDA results and so we report the "relaxed" results here and present results from the "strict" analysis in Supplementary Table 9. However, for analyses of plant richness (GAMs, Total DNA trend in Fig. 2), we used the "strict" filtered data set due to the sensitivity to outliers in these analyses.

In the 16S data set, we determined final taxonomic assignments based on scrutiny of NCBI BLAST[97] hits and comparison to known temporal biogeographic distributions (full justifications are given in Supplementary Data 3). Data for 16S sequences assigned to the same taxon were collapsed. We discarded all off-target and non-mammalian sequences, as well as those assigned as human (*Homo sapiens*) or wild boar (*Sus scrofa*), as the former is a common contaminant (also found in the negative controls) and the latter cannot be differentiated from sympatric domestic pig at this locus (Supplementary Data 3). Although absent from the negative controls, we note that domesticated taxa (pig, cow, sheep, goat) can be sources of sporadic PCR contamination[98]. Although we retained detections from all samples, we interpret sporadic occurrences of these taxa, defined as single, isolated PCR replicate detections, as likely deriving from contamination. We

note that these occur in otherwise low diversity samples that are most at risk of sporadic contamination detections (e.g., sheep at 6.1 ka; cow at 7.4, 9.1, -12.7 ka) (Fig. 3 and Supplementary Fig. 9).

We present a novel relative abundance index (RAI) to integrate the information of the relative abundance of reads and PCR replicability. It is a proportional index for each taxon within a sample and is calculated by multiplying the proportion of obtained reads by the proportion of weighted PCR replicates, the latter of which is a barcode detectability measure to account for differences in relative counts of reads across all retained barcodes within a sample (see also Rijal et al.[7]). This study follows the line of a growing number of environmental DNA modern studies that suggest that the number of DNA copies contain information about the species' relative abundance[99].

RAI = proportion of weighted PCR replicates x proportion of reads

## DNA sequences to identify plants associated with distinct elevational belts, and/or arable and pastoral activities

We used all the obtained sequences to get the maximum autecological information, even if not assigned at species level, for reconstructing distributional changes of plants typical for particular vegetation belts, and/or arable and pastoral activities. From all the obtained DNA sequences, only taxa with narrow elevational, climatic, and/or habitat requirements are good environmental indicators that can provide insights into past environmental change, while other sequences that do not originate from specific indicator taxa might smooth the results and by consequence, our palaeoecological interpretation. First, we matched the obtained sequences from Sulsseewli against the PhyloAlps database to obtain a list of the species that share the same sequence (haplotype-sharing species from now on). Then, we checked these haplotype-sharing species with the ecological parameters of Flora Indicativa[22] for temperature, which is the main parameter that characterises the elevational vegetation belts. The obtained results were graded from optimal to not suitable indicators. For example, if one sequence has 4 haplotype-sharing species and all of them have the same ecological value, it is considered an optimal indicator for a certain elevational belt, while conflicting values indicate a non-suitable indicator. Similarly, as for elevation, indicators for pastoral and arable activities were obtained from the sequence data based on the review of palaeoecological literature and Flora indicativa[22]. The regional coherence of all sequences was checked. Optimal indicator sequences for elevational vegetation belts and arable and pastoral activities as identified based on this procedure were plotted versus other palaeoecological indicators in Figs. 2 and 3. These comparisons formed the basis for interpretations regarding changes in vegetation caused by immigrations or expansions of plant species typical for various elevations belts or promoted by human activities. If climate was the main driver of past vegetation change, then changes in taxa representative of different vegetation belts are expected to track the chironomid temperature record. If human impact was the main driver, then these changes would be expected to follow the patterns of other palaeoecological indicators of human activities. Constrained incremental sum of squares (Coniss) zonation for all samples was used to infer statistically significant changes of plant taxa along the core.

## Statistics

For the statistical analyses, temperature, precipitation, charcoal, and coprophilous fungi spores were inferred for the depths that corresponded to the DNA samples using the nearest neighbour interpolation. We performed a fourth square-root transformation of the RAI of the species composition and the elevational stages (excluding the contaminants and the bad quality samples). We performed a Detrended Correspondence Analysis (DCA) to calculate the length of the environmental gradient in the composition data. Since the species composition and the elevational belts had lower than 3 SD axis length,

both linear and unimodal ordination methods can be applied to our data[100]. Thus, Redundancy Analysis (RDA) was performed with 12 ecological variables to explore the influence of climate (represented by chironomid-inferred temperatures, CHELSA-TraCE21k v1.0 model-inferred precipitation, organic matter content (LOI), wild mammalian sedaDNA) and human impacts (coprophilous fungi spores, sedaDNA from domesticated mammals, and microcharcoal). The ordistep function of vegan following the AIC criterion was used to select the most informative model, removing the environmental variables that were not informative (either did not explain significant variance in the RAI data or were highly correlated with variables that explain more variance). After obtaining the RDA on the optimal model, we checked the significance of both the RDA components and the individual explanatory variables using Pillai's trace statistic. In addition, we performed a piecewise redundancy analysis[101] to crosscheck the robustness of the RDA analyses of the effect of the 12 explanatory variables on the species composition (Supplementary Fig. 12A).

Given that the palaeoecological data rarely fulfils the assumptions of parametric tests, we used non-parametric generalised additive models (GAMs) based on "strict" filtered data to assess the relationship between plant richness and different explanatory variables. In addition, GAMs effectively capture nonlinear covariate effects generally expected in longer temporal data[102]. The plant richness was calculated as the total number of taxa recorded in each sample. We considered plant richness as the response, and chironomid inferred temperature, precipitation estimated from CHELSA data, coprophilous fungi influx, and the RAI proportion of different domesticated herbivores (cow, goat, and sheep) as the explanatory variables while fitting the GAMs. We used Poisson distribution with the log link to handle the count data[103]. A generalised linear model (GLM) was built with the same settings as of GAM and interrogated for the presence of potential outliers as residuals of the GAM models were not behaving properly. Sample EG33_L652 was detected as an outlier and removed from the final analysis, which improved the distribution of the residuals (Supplementary Fig. 14). Summary statistics for a full model with all the explanatory variables explained above, and a final model with covariates that had a statistically significant impact on plant richness are provided in the Supplementary Table 10. We also interrogated the final model to evaluate the presence of autocorrelation by calculating Durbin–Watson statistics. The negative autocorrelation (−0.32) at lag-1 was not statistically significant (D–W statistic = 2.62, $P > 0.05$). The model predictions provided in Fig. 5 were generated from the final model based on the 500 points of explanatory variables ranging from the minimum and maximum values of the observed data.

Furthermore, to test direct and indirect effects of climate, wild mammals and domesticates on richness, we performed a SEM by using the function gls on the "strict" filtered data with the piecewiseSEM package in $R$[104,105] (Supplementary Fig. 15). SEMs are built using a list of structured equations, which can be specified using the most common linear modelling functions in $r$, and thus can accommodate non-normal distributions, hierarchical structures and different estimation procedures. We extracted standardised model coefficients for each direct and indirect path, as well as marginal $R^2$ - the variance explained by fixed factors. The complete outcomes of the fitted model tested in the piecewiseSEM showing the standardised coefficients of precipitation, wild mammals and domesticates on richness are shown in Supplementary Table 11. Plots were made with R v3.4.2 using the Vegan, Rioja and Ggplot2 packages, whereas RDA and CONISS analyses were calculated in $R$ using vegan and rioja, respectively[106–108]. The GAM analyses were performed using mgcv[103] and the D–W statistic was calculated using the car packages in $R$[109].

## Reporting summary

Further information on research design is available in the Nature Research Reporting Summary linked to this article.

## Data availability

The raw DNA sequence data generated in this study have been deposited in the European Nucleotide Archive (ENA) under BioProject accession code PRJEB52290. The unfiltered OBItools output tsv files used in this study are available in the Dryad (datadryad.org, https://doi.org/10.5061/dryad.7wm37pvx5), pollen and charcoal data are available in the Neotoma database under the accession codes https://doi.org/10.21233/NM0Y-GM04 and https://doi.org/10.21233/J8P9-G487 (https://www.neotomadb.org). Source Data for Fig. 3 can be found in Supplementary Data 1, 3, 5, 12 and in the following repositories (https://doi.org/10.21233/NM0Y-GM04, https://doi.org/10.21233/J8P9-G487). Source Data for Figs. 4, 5, and 6 can be found in Supplementary Data 1 and 3.

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

## Acknowledgements

Special thanks to the Bergschaft Suls for coring permission, César Morales-Molino who helped with fieldwork, and for processing the pollen data for the Alpine Pollen Database (ALPADABA) and Neotoma, to Willi Tanner who also helped in fieldwork, Lucas Dane Elliott for doing part of the bioinformatic pipeline, Scarlett Zetter for preparing part of the metabarcoding libraries and Amy Mcdermott for analysing the LOI. Bioinformatic analyses were performed on resources provided by UNINETT Sigma2 - the National Infrastructure for High-Performance Computing and Data Storage in Norway. This study has been financed by the project "ECOGEN - Ecosystem change and species persistence over time: a genome-based approach", funded by the Research Council of Norway (grant number: 250963/F20 to Alsos). The PhyloAlps reference database was built thanks to the following projects: the joint ANR-SNF project Origin-Alps (ANR-16-CE93-0004, SNF-310030L_170059), European Research Council under the European Community's Seventh Framework Programme FP7/2007-2013 grant agreement 281422 (TEEMBIO) and by the SNF grant 31003A_149508. The sequencing for the PhyloAlps reference database was performed within the framework of the PhyloAlps project, funded by France Génomique (ANR-10-INBS-09-08). Some computations related to the elaboration of PhyloAlps data were carried out with the GRICAD infrastructure (https://gricad.univ-grenoble-alpes.fr).

## Author contributions

I.G.A. and S.L. designed the research, raised the funding, and provided resources; S.L., P.A.C. and E.C. generated and curated the PhyloAlps database; I.G.A., C.S., and F.R. did the fieldwork; S.G.P. did the ancient DNA laboratory work with input from I.G.A. and P.D.H.; T.G. performed radiocarbon dating; C.S. built composite cores and S.G.P. performed age-depth modelling with input from C.S. and O.H.; F.R. performed pollen, charcoal and non-pollen palynomorphs analysis; M.H. and A.R. performed chironomid analysis supervised by O.H., who also built the temperature reconstruction; D.N.K. and L.P. performed the precipitation time series; S.G.P. verified and curated the plant barcode sequence taxonomic assignments with input from

I.G.A. and J.P.T., who also verified their botanic origin; P.D.H. designed the blocking primers, verified the mammal barcode sequence taxonomic assignments, and performed the plant *sed*aDNA data quality control analyses; O.W. and S.G.P. designed the bioinformatic pipeline to obtain the indicator sequences, which were verified by J.P.T.; S.G.P. and O.W. developed the Relative Abundance Index; S.G.P. and Y.L. performed bioinformatics; S.G.P. and D.P.R. did the statistical analyses with input from O.W.; S.G.P. curated the data; W.T., I.G.A., S.L., C.S., J.P.T., A.G.B., F.R. and O.H. contributed with the palaeoecological and botanical interpretation; P.D.H. and K.W. provided interpretation of the mammalian data; K.W. provided archaeological interpretation of the region. S.G.P., I.G.A., and O.H. wrote the first draft of the manuscript, with contributions from J.P.T., P.D.H., W.T., C.S.,. L.P., D.P.R., and the remaining co authors commented on; all authors have reviewed and approved the final manuscript.

## Funding

## Competing interests
The authors declare no competing interests.

## Additional information

[1]The Arctic University Museum of Norway, UiT - The Arctic University of Norway, NO-9037 Tromsø, Norway. [2]Université Grenoble-Alpes, Université Savoie Mont Blanc, CNRS, LECA, 38000 Grenoble, Rhône-Alpes, France. [3]Palaeoecology, Institute of Plant Sciences & Oeschger Centre for Climate Change Research, University of Bern, 3013 Bern, Switzerland. [4]Fondation Aubert, 1938 Champex-Lac, Switzerland, Department of Plant Sciences, University of Geneva, 1292 Chambésy, Switzerland. [5]Norwegian College of Fishery Science, UiT - The Arctic University of Norway, Tromsø, Norway. [6]Department of Evolutionary Biology, Ecology and Environmental Sciences and Biodiversity Research Institute (IRBIO), University of Barcelona, Barcelona, Catalonia, Spain. [7]Department of Environmental Sciences, University of Basel, 4056 Basel, Switzerland. [8]Department of Archaeology, University of York, York 11 YO1 7EP, UK. [9]Faculty of Geographical and Geological Sciences, Adam Mickiewicz University, 61-680 Poznań, Poland. [10]Swiss Federal Research Institute for Forest, Snow, and Landscape Research (WSL), Zürcherstrasse 111, 8903 Birmensdorf, Switzerland. [11]Department of Environmental System Science, Institute of Terrestrial Ecosystems, ETH Zurich, Zürich, Switzerland. [12]Swiss Federal Research Institute WSL, Birmensdorf, Switzerland. [29]These authors jointly supervised this work: Oliver Heiri, Inger Greve Alsos. *A list of authors and their affiliations appears at the end of the paper. ✉e-mail: sandra.garces-pastor@uit.no

## The PhyloAlps Consortium

**Sébastien Lavergne[2], Eric Coissac ⓘ [2], Charles Pouchon[2], Cristina Roquet[2,13], Wilfried Thuiller[2], Niklaus E. Zimmermann[14], Adriana Alberti[15,16], Patrick Wincker[15], Martí Boleda[2], Frédéric Boyer[2], Anthony Hombiat[2], Christophe Perrier[17], Rolland Douzet[17], Jean-Gabriel Valay[17], Serge Aubert[17], France Denoeud[15], Bruno Bzeznick[2], Ludovic Gielly[2], Pierre Taberlet[2], Delphine Rioux[2], Céline Orvain[15], Maxime Rome[17], Rafael O. Wüest[14], Sonia Latzin[14], John Spillmann[14], Linda Feichtinger[14], Jérémie Van Es[18], Luc Garraud[18], Jean-Charles Villaret[18], Sylvain Abdulhak[18], Véronique Bonnet[18], Stéphanie Huc[18], Noémie Fort[18], Thomas Legland[18], Thomas Sanz[18], Gilles Pache[18], Alexis Mikolajczak[18], Virgile Noble[19], Henri Michaud[19], Benoît Offerhaus[19], Cédric Dentant[20], Pierre Salomez[20], Richard Bonet[20], Thierry Delahaye[21], Marie-France Leccia[22], Monique Perfus[22], Stefan Eggenberg[23], Adrian Möhl[23], Bogdan-Iuliu Hurdu[24], Paul-Marian Szatmari[24], Mihai Puşcaş[25], Jan Smyčka[2,16], Patrik Mráz[26], Kristýna Šemberová[26], Michał Ronikier[27] & Marek Slovák[28]**

[13]Systematics and Evolution of Vascular Plants (UAB) – Associated Unit to CSIC, Departament de Biologia Animal, Biologia Vegetal i Ecologia, Facultat de Biociències, Universitat Autònoma de Barcelona, ES-08193 Bellaterra, Spain. [14]Swiss Federal Research Institute WSL, CH-8903 Birmensdorf, Switzerland. [15]Génomique Métabolique, Genoscope, Institut François Jacob, CEA, CNRS, Université Evry, Université Paris-Saclay, FR-91057 Evry, France. [16]Université Paris-Saclay, CEA, CNRS, Institute for Integrative Biology of the Cell (I2BC), FR-91190 Gif-sur-Yvette, France. [17]CNRS, Lautaret, Jardin du Lautaret, Université Grenoble Alpes, FR-38000 Grenoble, France. [18]Conservatoire Botanique National Alpin, Domaine de Charance, FR-05000 Gap, France. [19]Conservatoire Botanique National Méditerranéen, FR-83400 Hyères, France. [20]Parc National des Ecrins, FR-05000 Gap, France. [21]Parc National de la Vanoise, FR-73000

Chambery, France. ²²Parc National du Mercantour, FR-06006 Nice Cedex 1, France. ²³Info-Flora – Centre national de données et d'informations sur la flore de Suisse, Genève CH-3001 Bern, Switzerland. ²⁴Institute of Biological Research, National Institute of Research and Development for Biological Sciences, RO-400015 Cluj-Napoca, Romania. ²⁵Department of Taxonomy and Ecology, Faculty of Biology and Geology and Al. Borza Botanic Garden - Babeș-Bolyai University, RO-400015 Cluj-Napoca, Romania. ²⁶Department of Botany, Faculty of Science, Charles University, CZ-12801 Prague, Czech Republic. ²⁷W. Szafer Institute of Botany, Polish Academy of Sciences, PL-31512 Kraków, Poland. ²⁸Department of Vascular Plant Taxonomy, Institute of Botany, SK-845 23 Bratislava, Slovak Republic.

