## [Peer Review File · Nature Communications]

Title: High resolution ancient sedimentary DNA shows that alpine plant diversity is associated with human land use and climate changeReviewers' comments:

Reviewer #1 (Remarks to the Author):

The Manuscript NCOMMS-21-36616 examines changes in vegetation from a Swiss alpine lake sediment core. Authors used a combination of classic paleoecological techniques and advanced DNA metabarcoding to reconstruct the environment during the last 12,000 years, including temperatures, plant species and human activity presence. Results indicate that vegetation composition changed since the begin of the Holocene. Former steppe-tundra vegetation was replaced by woodland and ultimately pastures. Authors conclude that governments should finance land use practices.

The study makes use of high-throughput sequencing techniques to reconstruct past environments. Although not completely new (see <https://www.mdpi.com/2571-550X/4/1/6> for a review), ancient eDNA sequencing is definitively on the rise (<https://www.nature.com/articles/s41586-021-04016-x>). This methodological advance is definitively the most novel and noteworthy part of the manuscript. This allowed the identification of a fairly high number of plant species as compared to classic palynological techniques. Nevertheless, admittedly, the overall picture stays the same. Qualitatively and conceptually wise, there is no noteworthy breakthrough. In fact, as also noted by the same authors, the vegetation pattern looks similar to other previously-studied alpine lakes.

Furthermore, there are two main problems: the strength of data analysis and the reliability of deductive conclusions.

The whole report of vegetation pattern change (lines 163-313) relies on a single redundancy analysis (RDA). This is quite poor and insufficient for supporting a four-pages result section. First, RDA requires linear relationships between variables. Clearly, the data examined here are not linear, thus violating a necessary assumption, impacting the reliability of results. Second, the significance of those relationships was not provided, thus impeding to gather whether those associations are significant or not. Third, this descriptive analysis does not support any claim regarding the effects of environmental variables on vegetation.

Most importantly, authors continuously claim that plant diversity increase or decrease in response to temperature and land use. Unfortunately, there is no evidence in support of those conclusions. Trends of plant diversity are not analyzed at all, why? Indeed, there is no evidence of any quantitative kind in support of authors' conclusions. Given the complexity of the data and the bold claims authors made, a much deeper and robust inference analysis is necessary. Mixed models or structural equation models are necessary to properly infer the effects of temperature and other variables on plant diversity. Given that data are spatiotemporally correlated, a proper consideration of random effects is necessary too. Only such kind of hierarchical modelling can support authors' claims.

Furthermore, I was desperately looking for the figure reporting plant diversity change patterns, but such figure is unfortunately hidden at the end of SI (instead, a classic but trivial picture of vegetation belt is included as main fig. 1), why?

In fact, just looking and inspecting the figure, one can notice that authors conclusions are actually not supported by data. Authors claimed that human agropastoral actives during 3.5 ka and late Roman period (1.6 ka) increased plant diversity. Unfortunately, one can see that: (1) plant diversity is

continuously oscillating, and (2) on average, plant diversity is not higher during that time but rather quite lower as compared to other time periods, such as the Early Holocene (11 ka).

Additional nontrivial problems include:

- Not clear what the knowledge gap is.
- Relative abundance: How was the effect of differential DNA degradation accounted for? How was the impact of sequencing depth accounted for? How did you take into account the efficiency of the tax-identification pipeline among different taxa?
- Claims around taxa found at different altitudes. Those are just generalist taxa with a broad altitude distribution, which is quite common for alpine plants *sensu lato*. So, being distributed at different altitudes, they also co-occur at a given, specific point within their normal range. Associated to that, as for normality definition provided at lines 123, authors definition of normality sounds weird. What is normal? How do you define normality?
- Traditional activities. What is meant by "traditional" in this manuscript? There is a huge time (hence technological) gap between the type and impact of human activities pertinent to thousands of years ago and current 'traditional' activities. The two are just incomparable. Please avoid using ambiguous terminology and comparing apples with oranges.
- Keep in mind the strong limitations of such kind of data for inferring ecological processes. Given the nature of sedimentary data, you cannot infer fundamental biotic processes responsible for driving biodiversity, including very-well known ecological factors at individual (e.g. physiological, fitness), local (e.g. species interactions, mortality), and regional (e.g. dispersal and habitat connectivity) scales.
- Climatic variables associated with precipitation and humidity known to play a fundamental role in alpine vegetation were unfortunately ignored.
- How was the integrity of the sedimentary sequence assessed? Results at lines 206–208 are odd, as *Poaceae* and *Artemisia* are often associated with cold and dry climate, while here it is stated that they increase with increasing temperature. Furthermore, there is no result associated with humidity to help resolving such issue.
- Confusion around trees and alpine belt. The alpine belt is, by definition, treeless (see C. Körner *Alpine Plant Life*). The treeline is a climatic limit; notably, this does not have to be confused with the timberline. If natural reforestation is occurring, then trees are just going back to their original place. This means that such vegetation belt is not alpine.
- Conclusions: First, it is so odd to still find scientist supporting livestock and associated policy given the already-well-consolidated evidence pointing out the contrary, i.e., the negative impact of livestock on biodiversity and ecosystems (e.g. <https://www.pnas.org/content/110/52/20900>; <https://iopscience.iop.org/article/10.1088/1748-9326/10/11/115004/meta>). Second, I would refrain to get to such political conclusion given the very narrow results of this study. Please stay in the realm of science and keep your political/ideological views out of this context. Notably and ironically, the Swiss government actually already DOES finance farmers and the agropastoral business (<https://sustainabledevelopment.un.org/index.php?page=view&nr=685&type=504&menu=139>; <https://www.oecd-ilibrary.org/sites/05024436-en/index.html?itemId=/content/component/05024436-en>). As a matter of facts, such policy is highly debated and controversial (e.g. https://www.swissinfo.ch/eng/business/agriculture_the-privileges-of-being-a-farmer-in-switzerland/43613212).

Reviewer #2 (Remarks to the Author):

This paper reconstructed past vegetation, climate, and livestock over the past ~12,000 years at the Lake Sulsseewli (Alps) by using several approaches: sedimentary ancient DNA (sedaDNA), pollen, spores, chironomids, and microcharcoal. The authors built highly-complete local DNA reference library (from a previous project) and used this to obtain a quite rich sedaDNA local database (366 plant taxa). They found that vegetation mainly depended on temperature during the first half of the Holocene, while human activity drove changes from 6 ka onwards. Land-use shifted from episodic during Neolithic to agropastoral intensification in Medieval Age.

Overall, I found the manuscript really interesting even if the main message and the approach are moderately novel. Therefore, I think the manuscript could be ameliorated in the part of background and data interpretation. In particular, I would focus on how the multiproxy and high resolution approach could improve the interpretation of classical biogeographic theories like those inherent to glacial and interglacial refugia also in relation to treeline dynamics.

Main remarks

- 1) The paper is well-organized and well-presented but the methodology is relatively novel. Numerous publications already exist on sedaDNA (Eg. Zimmerman et al. 2017; Sisi Liu et al. 2020 and 2021 and other), despite this work uses a multiproxy and high resolution approach (see Parducci et al. 2015; ter Schure et al. 2021). Of course this allowed to authors to identify 366 taxa but this is no surprising to me: more sampling effort using a regional reference database and implementation with different approaches (pollen, microcharcoal, etc.), imply more records.
- 2) I found that the main scientific message it brings in discussion and conclusion (i.e. "plant biodiversity is a result of human land use") is, in a certain way, already seen or heard. Other authors have reached the same conclusions with more conservative and (maybe) coarse but less expensive methods (i.e. pollen analyses). In addition, the main conclusions are based on a single site (Lake Sulsseewli) so we cannot know if (for instance) in sites at lower elevation human impact started earlier or in in less favourable areas the human influence was less marked (see Rösch et al. 2021).
- 3) In the background and discussion, the likely treeline changes during time (in the picture within Fig. 1 we can observe treeline...) is narrowly treated. However, SedaDNA and pollen have different but complementary abilities for reconstructing past vegetation including treeline dynamics also in relation to human presence. The discussion paragraphs from line 330 to 388 could be improved in this sense.
- 4) Nothing is said about the spreading of taxa from various Glacial refugia, in response to climate changes during the Holocene. E.g. Can the multiproxy and high resolution approach disclose more information about glacial and interglacial Holocene refugia also considering the future perspective for the area (e.g. permafrost degradation, change of alpine slope dynamics)?
- 5) In the discussion, I do not find any mention regarding the Little ICE Age (LIA). Indeed, from Fig. 2 we can observe an increase of Subalpine and Alpine species during the last 1000 years.

Other points

L71: ... and forms elevational belts ◊ weird sentence. I suggest to replace with "...determined by temperature, landforms and geomorphological processes along elevational belts". ...or similar sentence. Indeed, not only landforms influence vegetation but also the intensity of processes acting along slopes

L74: not clear to what refers the term “present-day” in relation to alpine species. Considering the concept of “alpine life zone” alpine species can change the optimum elevation over time. So, I would remove “present-day”

L93: I suggest to replace immigration with “migration” since the word immigration implies only movement from the external regions to the Alpine area and not for instance internal movements from an alpine area to another one. I know that in zoology the term migration refers to seasonal movement of animals but this meaning do not apply for plants or humans.

L163-185. Didn’t you perform any statistic test on the RDA analysis? E.g. variable selection, significant variables, variance etc. I think that to understand the most important variables is an added value

Reviewer #3 (Remarks to the Author):

Review of Garcés-Pastor et al., *Nature Communications*
October 2021

Summary

Garcés-Pastor et al. present a multi-proxy lake sediment record from the Swiss Alps that provides detailed insight into climate- and human-driven ecological change over the last ~12,500 years. The sedaDNA datasets (plant and mammal) are particularly rich and yield nuanced records of plant community composition and human activity that would not be possible with traditional analyses. The authors employ a couple of new approaches to interpret the sedaDNA data, including identifying local elevation band and pastoral/arable indicator taxa and introducing a useful metric, the “relative abundance index.” Their inferences about human impact are corroborated by traditional analyses of microcharcoal and coprophilous fungal spores. While the data presented are robust and yield interesting insights about human impacts in the Alps, the manuscript is currently a bit challenging to follow because of some missing context/results and a substantial amount of interpretation included prematurely in the results section. Overall, I believe the paper will make a nice contribution to *Nature Communications*, but it would benefit from substantial revision of the text and minor updates to most of the figures. Below I summarize my major comments and line-by-line suggestions.

Major comments

- **Background and context of land-use:** Given that much of the focus of this paper is on the impact of human activities, more information on the recent/historical human occupation of this area (catchment and/or region of the Alps) is needed to place the proxy records in context. Does the timing and style of land use suggested by your dataset fit with what was already known about human activity here? Are there other proxy records from nearby sites or similar elevations in the Alps that corroborate this history? I recommend this be added either to the end of the introduction or a new first subsection within Results, with a brief comparison with other regional proxy records included in the Discussion.
- **Results/Discussion structure:** The manuscript is currently lacking a clear first-order description of much of the data (including the core lithostratigraphy/age model and the proxy datasets), and the

results section contains quite a bit of interpretation (e.g., inferring the drivers of vegetation change) that would better fit into the discussion. As such, it's hard to follow the results section because it does not walk the reader through the results before diving into the more nuanced second-order takeaways. I suggest reorganizing the text to include a brief results subsection for each proxy (or group of proxies); then, the discussion could include a narrative with interpretations for each time period (as is currently the organization for the results section, but ideally more concise). Alternatively, this could evolve into a "result and discussion" section, but I think that would be harder to follow without first being introduced to the general proxy results/patterns as part of a strictly "results" section.

- **Interpretations about temperature driving plant richness change:** The authors suggest that temperature increases during the early part of the record result in increased plant diversity and contrast this finding with Liu et al., 2021. However, this may be an overinterpretation of the RDA results; the time series of chironomid-inferred temperatures and sedaDNA-inferred plant richness (Fig. 2) shows that this relationship is nuanced and that the most prominent warming phase (~11 to 9 ka) corresponds to a steady decrease in richness, which supports the findings of Liu et al. I suggest that this interpretation be re-evaluated, at least with text changes but also perhaps with a more direct assessment of the correlation between temperature and richness.

Line-by-line comments

Line 52: "Biodiversity hotspot" has a specific definition based on number of endemics and threatened species; I'm not sure it's appropriate to apply it generally to "alpine areas." Consider rephrasing.

Lines 60-61: Phrasing a bit awkward; consider changing to "more intensive agropastoralism" or something similar.

Line 77: I recommend splitting this into two paragraphs before "Plant remains..." with some sort of transition.

Lines 89-90: No need for quotes around "natural" and "human."

Line 103: Change "composition" to "richness" (assuming that's what is meant).

Lines 105+: Somewhere, it would be useful to include a bit of information about the historic/recent land use practices in this area to provide context.

Lines 114-126: This paragraph contains a lot of results for the introduction; it currently reads more like an extended abstract. I recommend revising to remove some of the results sentences and keep this more as a "roadmap" paragraph that introduces the study approach.

Line 128: It would be helpful to include a brief results section focused on the sediment core lithology and chronology, rather than jumping right to the proxy records. Some details about the lithologic changes and age model (including the notable increase in sedimentation rate around 5 ka) are important to understanding this record and should be mentioned at least briefly in the main text.

Lines 158-160: Are the temperatures in parentheses means for the time period specified? If so, describe them as such. If not, it would probably be better to round to the nearest degree and give a range for that time period, as done on line 197.

Line 162: Before explaining the drivers of vegetation changes, there needs to be brief results sections describing all the proxy results (sedaDNA of plants and mammals, pollen, charcoal, coprophilous fungi). Determining the drivers of change is a second-order result that probably belongs in the discussion.

Line 169: I believe Fig. 3 has not yet been referenced (in part, because there is currently no results section describing much of the proxy data).

Line 175-176: This type of interpretation belongs in the discussion, not the results section.

Line 189-190: The base of the core definitely seems to be older than the end of the YD (11.7 ka) based on the age model. I would rephrase to say something like "likely originate from around the Younger Dryas event."

Line 195: Reference Fig. 2 when describing the sedaDNA results.

Line 197: There is no panel F or G on Fig. 2.

Line 199-200: This is also interpretation that belongs in discussion. Also, temperature looks to be about the same (probably within error) in Zones 1 and 2 (as they are outlined in this figure).

Line 203: Figure reference should be SF8.

Lines 203-206: More interpretation that belongs in discussion, here and in subsequent paragraphs.

Lines 235-236: The fungal spores record shows similar flux values in the earliest part of the record, so I'm not sure I agree with this description/interpretation.

Lines 244-245: To my eye, there is not much happening at 6.35 ka; it looks comparable to "background" levels of human indicators from first half of the record. There is no discernible change in microcharcoal. If you would like to point to an increase in grazing indicators, I recommend at least reducing the temporal precision ("around 6 ka", or simply remove that and leave only "during the latter half of the Neolithic (7.5-4.2 ka)") because this is not a very sharp transition. Also, move the Fig. 3 reference to the end of the sentence (with Fig. 2).

Line 254: Does the appearance of red deer suggest human impact? Please explain this connection more clearly.

Line 320: Perhaps this should be "global **or** incomplete **regional** reference databases"?

Line 344: For clarity, I would just change "p6 loop sequence" to something more general like "marker sequence" or "barcode sequence" for readers less familiar with these methods.

Line 347-348: This sentence is a bit vague; perhaps give an example of how this indicator sequence approach could be applied?

Lines 354-356: The highest richness of the early part of the record (~11-10 ka) precedes the major temperature rise and actually coincides with quite low temperatures. The RDA does point to climate being a driver of composition change during the first half of the record, but I'm not sure this statement about temperature and plant diversity specifically is supported by the data.

Lines 357-359: More accurately, Liu et al. reported a **negative** relationship between temperature and plant richness. And actually, I think there is similarity between the records when looking at the alpine/subalpine-dominated interval of the Sulsseewli record (up to ~9 or 9.5 ka). Both show relatively high richness corresponding to low temperatures, followed by a decrease in richness when temperature rises. It would be worth exploring this relationship more directly, and if the relationship changes once montane/colline vegetation is more abundant, that's an interesting result.

Lines 360-361: I'm not convinced that different ID thresholds means that the two richness records aren't comparable, particularly when both studies are focusing on relative change through time (not comparing absolute richness).

Line 363-365: Can you rule out that there is no transport of plant DNA from higher elevations down into the lake, given excellent preservation, high overall sed rates, and high clastic flux (potentially promoting more stable transport of mineral-bound DNA)? It would be prudent to point out this possibility at least. I believe that most modern sedDNA validation studies (e.g., Alsos et al., 2018) have been conducted in lower-relief settings, so it's hard to rule out enhanced DNA transport in this setting.

Lines 389-390: This topic sentence is very similar to that of the previous paragraph; modify to reflect that this paragraph is more about high levels of richness being associated with human activity.

Line 392: "Millennia" should be "millennium"

Lines 399-400: It's unclear what impacts you're referring to about here. Reduction in richness? Also, if you are comparing with Liu et al specifically (as indicated in next sentence), say that instead of "studies elsewhere," which is vague.

Line 402: It's a bit unusual to say "our findings stand with"; I recommend changing to "our findings support."

Line 405: I suggest briefly explaining "intensive" vs "extensive" in this context, as your readership may not be familiar.

Lines 410-412: It's confusing how the observation from the previous sentence is "in line with" studies looking at the relationship between richness and temperature, given that the previous sentence is about human land-use, not climate change. Please clarify/revise.

Line 417: The phrasing "the within dominant alpine families" is unclear; please rephrase.

Lines 418+: The flow of logic through this paragraph is difficult to follow. The previous sentence is about more precise metrics for tracking/documenting alpine biodiversity change. The transition "Therefore" suggests the next idea stems from that, but it is instead about promoting traditional land-use practices to conserve biodiversity, which is quite different. The transition between these sentences should be changed, and perhaps the more prescriptive conclusions about maintaining traditional land-use should be in its own paragraph.

Lines 423-424: It's rather unusual to bring up new ideas (mammalian impacts) in the final sentence of the manuscript main text. Consider adding a final concluding/forward-looking sentence.

Lines 431-432: Are there any outlets, or is it a terminal lake?

Lines 432-434: It would be more relevant to provide local climate parameters if possible, rather than lowland values.

Lines 445-446: It's very difficult to discern these lithostratigraphic markers in Fig. SF1. Please increase the resolution of this figure and the text annotations. What is the difference between the red lines (correlation points) and blue lines (lithostratigraphic features)? More information is needed on how the cores were correlated.

Lines 467-471: This currently reads as though the regression model/temperature calibration was newly generated for this study, which I don't think is the case. Please rephrase slightly to make it clear you used a previously published and broadly applied transfer function.

Line 510: "Generic" is a bit confusing; perhaps "universal" is better.

Line 561: Are the common lab contaminants that you removed included in a table? It's not clear if these are included in table ST6.

Lines 583+: This is a useful new metric that may be applied in future studies (I assume you hope it will be!). As such, it would be very useful to more clearly define RAI using an equation, as it's currently hard to follow this description. This should include the definition of weighted PCR replicates, rather than referring the reader to another study.

Figures:

Fig. 1: Label Switzerland. Specify in the caption that the lake is circled in red.

Fig. 2: In the caption, clarify what smoothing function is used in panel A. For panel D, add a label next to "D" for symmetry with the others (perhaps "Climate proxies"?). Somehow (with color?) make it clear which y-axis corresponds to LOI vs. July temp. Add "(%)" for LOI. Add standard error for air T estimates.

Fig. 3: Figure caption needs a description of the top panel (sedaDNA results binned by elevation band indicators). Reorder figure caption information to reflect the order in the figure itself (e.g., microcharcoal, arable plant sedaDNA, etc). It would probably make more sense to switch the positions of coprophilous fungi and pastoral sedaDNA so that the mammal indicators (and plant sedaDNA datasets) are grouped.

Fig. 4: Please explain what the "Groups" are – less than that age? It would be clearer to list the age range (e.g., <3 ka, 3–6 ka, etc.) included for each category, and replace "Groups" with "Age group". Is Colline belt not on the plot because it is beyond the limits of this plot? If so, perhaps draw an arrow pointing toward where it would plot. If not, explain more clearly what you mean here.

Table ST1. Please add a column with median calibrated age (or 95% range) so this information can more easily be compared with the age-depth model.

Fig. SF5: Define "LPAZ."

Table ST2: The “percentage” column appears empty – are values missing here? If so, please explain in the caption what this percentage refers to. If not, remove.

Table ST5: I could not find a comparable table for arable indicators. Perhaps these could be incorporated into the same table, structured more like Table ST2.

Response to reviewers

Dear reviewers,

We thank you very much for your nice comments. We think that they have contributed to considerably improve the quality of the manuscript.

After performing the additional statistical analyses suggested by the reviewers and adding new precipitation data to our multiproxy approach, results draw us to the same conclusions with more robust statistics.

The authors have addressed all the reviewers' suggestions, which have improved the structure and content of the manuscript, and we respond to them point-by-point below.

Yours,

Sandra Garcés-Pastor and co-authors

Updated figures and tables:

Revised manuscript

Figure: 2, 3, 4, 5

Supplementary Figure: SF1, S10, S11, S12, S13

Supplementary Table: ST1, ST2, ST5

Supplementary dataset: S1 v2, S6

Reviewer #1 (Remarks to the Author):

The Manuscript NCOMMS-21-36616 examines changes in vegetation from a Swiss alpine lake sediment core. Authors used a combination of classic paleoecological techniques and advanced DNA metabarcoding to reconstruct the environment during the last 12,000 years, including temperatures, plant species and human activity presence. Results indicate that vegetation composition changed since the begin of the Holocene. Former steppe-tundra vegetation was replaced by woodland and ultimately pastures. Authors conclude that governments should finance land use practices.

The study makes use of high-throughput sequencing techniques to reconstruct past environments. Although not completely new (see <https://www.mdpi.com/2571-550X/4/1/6> for a review), ancient eDNA sequencing is definitively on the rise (<https://www.nature.com/articles/s41586-021-04016-x>). This methodological advance is definitively the most novel and noteworthy part of the manuscript. This allowed the identification of a fairly high number of plant species as compared to classic palynological techniques. Nevertheless, admittedly, the overall picture stays the same. Qualitatively and conceptually wise, there is no noteworthy breakthrough. In fact, as also noted by the same authors, the vegetation pattern looks similar to other previously-studied alpine lakes.

Our comment: The similarity between our DNA based reconstruction and previous pollen-based studies validates our molecular approach. However, *seadNA* gives a higher-resolution insight especially about the forb communities, which are usually underrepresented by pollen records. The novelty lies in the much more detailed floristic and ecological reconstruction, which allows detecting both responses to climate and humans.

Furthermore, there are two main problems: the strength of data analysis and the reliability of deductive conclusions.

The whole report of vegetation pattern change (lines 163-313) relies on a single redundancy analysis (RDA). This is quite poor and insufficient for supporting a four-pages result section. First, RDA requires linear relationships between variables. Clearly, the data examined here are not linear, thus violating a necessary assumption, impacting the reliability of results. Second, the significance of those relationships was not provided, thus impeding to gather whether those associations are significant or not. Third, this descriptive analysis does not support any claim regarding the effects of environmental variables on vegetation.

Our comment: Thanks for pointing out these shortcomings. Our data are indeed linear, but we forgot to report that.

Action taken: to support our claim, we selected the most informative model, removing the environmental variables that were not informative, following the AIC criterion. These new RDA models are added in Figure 4 and text is added in material and methods. The significant values in the Supplementary data S6.

Most importantly, authors continuously claim that plant diversity increases or decreases in response to temperature and land use. Unfortunately, there is no evidence in support of those conclusions.

Our comment: We thank the reviewer for reporting this point so we can address it. Indeed, the richness is poorly related to climate but strongly to human land use.

Action taken: A new graph was provided (Figure 5) showing the correlation between richness and RDA axis 1 and 2. Further, significant values provided in Supplementary data S6. Text was included in material and methods and results sections to update with these changes.

Trends of plant diversity are not analyzed at all, why? Indeed, there is no evidence of any quantitative kind in support of authors' conclusions. Given the complexity of the data and the bold claims authors made, a much deeper and robust inference analysis is necessary. Mixed models or structural equation models are necessary to properly infer the effects of temperature and other variables on plant diversity. Given that data are spatiotemporally correlated, a proper consideration of random effects is necessary too. Only such kind of hierarchical modelling can support authors' claims.

Our comment: We thank the reviewer for pointing out his statistical concerns.

Action taken: The *ordistep* function of *vegan* following the AIC criterion was used to select the most informative model, removing the environmental variables that were not

informative for the RDA analyses. The new optimal RDA analyses models were performed and included in Figure 4. We also performed a more complex GAM model but since it gave similar results, we kept only the RDA in the manuscript.

Furthermore, I was desperately looking for the figure reporting plant diversity change patterns, but such figure is unfortunately hidden at the end of SI (instead, a classic but trivial picture of vegetation belt is included as main fig. 1), why?

Our comment: We depicted the vegetation belts in Figure 1 to ensure that readers are familiar with them.

Action taken: The plant diversity change curve is now presented in Figure 2.

In fact, just looking and inspecting the figure, one can notice that authors conclusions are actually not supported by data. Authors claimed that human agropastoral activities during 3.5 ka and late Roman period (1.6 ka) increased plant diversity. Unfortunately, one can see that: (1) plant diversity is continuously oscillating, and (2) on average, plant diversity is not higher during that time but rather quite lower as compared to other time periods, such as the Early Holocene (11 ka).

Our comment: We realize that it is hard to see the correlation between diversity and human versus climate impact, especially as indeed the first colonization during the Early Holocene is climate driven.

Action taken: We added a new figure (Figure 5, see above) that shows the much stronger correlation between richness and RDA axis 1 (human related) than richness and RDA axis 2 (climate related). The text has been modified accordingly.

Additional nontrivial problems include:

- Not clear what the knowledge gap is.

Our comment: We thank the reviewer for pointing this out so we can make this information clearer in the introduction.

Action taken: The introduction was modified to highlight the knowledge gap (lines: 82-88).

- Relative abundance: How was the effect of differential DNA degradation accounted for? How was the impact of sequencing depth accounted for? How did you take into account the efficiency of the taxa-identification pipeline among different taxa?

Our comment: We thank the reviewer for pointing out his DNA quality concerns.

Action taken: We now provided a set of quality controls of all samples. This is added in Supplementary Figures SF10 and SF11 in Supplementary material, and text was added in the Material and Methods section.

- Claims around taxa found at different altitudes. Those are just generalist taxa with a broad altitude distribution, which is quite common for alpine plants sensu lato. So, being distributed at different altitudes, they also co-occur at a given, specific point within their

normal range. Associated to that, as for normality definition provided at lines 123, authors definition of normality sounds weird. What is normal? How do you define normality?

Our comment: We agree with the reviewer that the majority of taxa have a broad distribution. However, taxa without indication value (generalists) were excluded from our analysis, and only taxa that had a restricted altitudinal range were chosen. This information has been obtained from the Flora Indicativa, which is a reference book for the flora of the Alps and has the studied parameters for all its flora (Landolt et al. 2010). Furthermore, the botanical expert and coauthor, Jean-Paul Theurillat, has checked the botanical coherence and ecological indicative values of such taxa in the study site.

Action taken: The word “normally” was changed to “generally” (line: 127). We also rephrased part of the Materials and Methods section to better explain how the indicators were obtained (lines: 813-819).

- Traditional activities. What is meant by "traditional" in this manuscript? There is a huge time (hence technological) gap between the type and impact of human activities pertinent to thousands of years ago and current 'traditional' activities. The two are just incomparable. Please avoid using ambiguous terminology and comparing apples with oranges.

Our comment: We agree with the reviewer that this word can lead to misunderstanding.

Action taken: “traditional” has been deleted from the text as suggested.

- Keep in mind the strong limitations of such kind of data for inferring ecological processes. Given the nature of sedimentary data, you cannot infer fundamental biotic processes responsible for driving biodiversity, including very-well known ecological factors at individual (e.g. physiological, fitness), local (e.g. species interactions, mortality), and regional (e.g. dispersal and habitat connectivity) scales.

Our comment: We agree with the reviewer that this kind of data has limitations.

Action taken: We have improved our statistical analyses to better understand the effect of human and climate in long term vegetation changes. The graphs of that analyses are in Figure 4 and 5 and text was adjusted accordingly.

- Climatic variables associated with precipitation and humidity known to play a fundamental role in alpine vegetation were unfortunately ignored.

Our comment: We agree with reviewer 1 that other climatic variables beyond temperature can play an important role in Alpine vegetation. To address these concerns, we incorporated precipitation data obtained from the CHELSA-TraCE21k v1.0 model (Karger et al 2021) in our statistical analyses.

Action taken: We incorporated precipitation data to our statistics, added in Figures 2 and 4, and text was adjusted accordingly.

- How was the integrity of the sedimentary sequence assessed? Results at lines 206–208 are odd, as Poaceae and Artemisia are often associated with cold and dry climate, while

here it is stated that they increase with increasing temperature. Furthermore, there is no result associated with humidity to help resolving such issue.

Our comment: We thank the reviewer for pointing out these issues. We realised that the text should be improved for that section. We also look for data that could resolve the humidity issue that the reviewer suggests.

Action taken: we have rephrased the Material and Methods section to address the integrity of the sedimentary sequence (lines: 466-482), and changed the Supplementary Figure SF1 accordingly. Precipitation data was added to our statistical analyses and text to resolve the humidity issue.

- Confusion around trees and alpine belt. The alpine belt is, by definition, treeless (see C. Körner Alpine Plant Life). The treeline is a climatic limit; notably, this does not have to be confused with the timberline. If natural reforestation is occurring, then trees are just going back to their original place. This means that such vegetation belt is not alpine.

Our comment: We totally agree and that is why we also illustrated the alpine belts in Figure 1. We realise that our text has not been clear enough.

Action taken: we changed “Alpine areas” in the abstract to “The European Alps” in order to clarify the confusion around trees and alpine belt (line: 92).

- Conclusions: First, it is so odd to still find scientist supporting livestock and associated policy given the already-well-consolidated evidence pointing out the contrary, i.e., the negative impact of livestock on biodiversity and ecosystems (e.g. <https://www.pnas.org/content/110/52/20900>; <https://iopscience.iop.org/article/10.1088/1748-9326/10/11/115004/meta>). Second, I would refrain to get to such political conclusion given the very narrow results of this study. Please stay in the realm of science and keep your political/ideological views out of this context. Notably and ironically, the Swiss government actually already DOES finance farmers and the agropastoral business (<https://sustainabledevelopment.un.org/index.php?page=view&nr=685&type=504&menu=139>; <https://www.oecd-ilibrary.org/sites/05024436-en/index.html?itemId=/content/component/05024436-en>). As a matter of facts, such policy is highly debated and controversial (e.g.

https://www.swissinfo.ch/eng/business/agriculture_the-privileges-of-being-a-farmer-in-switzerland/43613212).

Our comment: Thanks for making us aware of these very useful papers and web pages. We agree that our statement was phrased too universal.

Action taken: we changed the last part of the abstract (lines: 70-71) and the discussion (lines: 443-446) as suggested by the reviewer.

Reviewer #2 (Remarks to the Author):

This paper reconstructed past vegetation, climate, and livestock over the past ~12,000 years at the Lake Sulsseewli (Alps) by using several approaches: sedimentary ancient DNA (sedaDNA), pollen, spores, chironomids, and microcharcoal. The authors built highly-complete local DNA reference library (from a previous project) and used this to obtain a quite rich sedaDNA local database (366 plant taxa). They found that vegetation mainly depended on temperature during the first half of the Holocene, while human activity drove changes from 6 ka onwards. Land-use shifted from episodic during Neolithic to agropastoral intensification in Medieval Age.

Overall, I found the manuscript really interesting even if the main message and the approach are moderately novel. Therefore, I think the manuscript could be ameliorated in the part of background and data interpretation. In particular, I would focus on how the multiproxy and high resolution approach could improve the interpretation of classical biogeographic theories like those inherent to glacial and interglacial refugia also in relation to treeline dynamics.

Our comment: We thank the reviewer for his comment. The ideas about exploring the classical biogeographic theories like vegetation refugia and treeline dynamics are very interesting. Unfortunately, with this single lake study we cannot properly explore such ideas, but as more high resolution data becomes available this will indeed increase our knowledge about the subject. Especially, such data would be needed from outside the major glaciers, as is currently available from only one LGM refugia in the north (Alsos et al. 2020 QRS). The treeline dynamics, on the other hand, can be explored with the current dataset.

Action taken: We added several words referring to treeline changes and deforestation in the discussion (lines: 397-399; 412).

Main remarks

1) The paper is well-organized and well-presented but the methodology is relatively novel. Numerous publications already exist on sedaDNA (Eg. Zimmerman et al. 2017; Sisi Liu et al. 2020 and 2021 and other), despite this work uses a multiproxy and high resolution approach (see Parducci et al. 2015; ter Schure et al. 2021). Of course this allowed the authors to identify 366 taxa but this is not surprising to me: more sampling effort using a regional reference database and implementation with different approaches (pollen, microcharcoal, etc.), imply more records.

Our comment: We agree with the reviewer that there are other publications with sedaDNA. However, our multiproxy approach provides the most detailed reconstruction of the Holocene to date. We also apply a novel method to obtain ecological indicators from the sequences and release the most complete *TrnL* p6 loop database for the Alps.

Action taken: We adjusted parts of text in the introduction to highlight the novelty of our study compared to the other sedaDNA studies.

2) I found that the main scientific message it brings in discussion and conclusion (i.e. "plant biodiversity is a result of human land use") is, in a certain way, already seen or heard.

Other authors have reached the same conclusions with more conservative and (maybe) coarse but less expensive methods (i.e. pollen analyses). In addition, the main conclusions are based on a single site (Lake Sulsseewli) so we cannot know if (for instance) in sites at lower elevation human impact started earlier or if in less favourable areas the human influence was less marked (see Rösch et al. 2021).

Our comment: We agree with the reviewer that plant biodiversity has been addressed using pollen, a proxy that also has its limitations. However, *sedDNA* offers a more accurate assessment of the magnitude of diversity change and, in our study, highlights the main role of forbs in shaping plant diversity of Sulsseewli, which is something new. This novelty can only be reached by *sedDNA*. Figure 2 A shows the richness of pollen and total *sedDNA*. Even though pollen and DNA might show similar general vegetation changes, pollen underestimates the richness that occurs in the region of the lake because it is biased by trees (see also supplementary figure SF7). This is now shown in Figure 2.

We agree with the reviewer that this is a study of the vegetation of the catchment of one lake and we cannot extrapolate our results to other lakes. A following study with more lakes is coming to study the human influence of the Alps vegetation.

Action taken: We added the richness curve to Figure 2. We rephrased part of the discussion to note that our conclusions are applied to alpine systems similar to Sulsseewli (lines: 433-434).

3) In the background and discussion, the likely treeline changes during time (in the picture within Fig. 1 we can observe treeline...) is narrowly treated. However, *SedaDNA* and pollen have different but complementary abilities for reconstructing past vegetation including treeline dynamics also in relation to human presence. The discussion paragraphs from line 330 to 388 could be improved in this sense.

Our comment: We thank the reviewer for reporting that, we added it to the discussion.

Action taken: We added several words referring to treeline changes and deforestation in the discussion (lines: 397-399; 412).

4) Nothing is said about the spreading of taxa from various Glacial refugia, in response to climate changes during the Holocene. E.g. Can the multiproxy and high resolution approach disclose more information about glacial and interglacial Holocene refugia also considering the future perspective for the area (e.g. permafrost degradation, change of alpine slope dynamics)?

Our comment: as reviewer 2 pointed out in former comments, this study is based on a single lake. In order to study the main migration pathways of species we would need several sites to check the patterns of the species distribution and/or study the intraspecific genetic variation. Even though we find the idea proposed by reviewer 2 very interesting, we cannot do such analyses with only one lake and a marker that has not enough intraspecific variation for all the vascular plants. Such study falls beyond the scope of the present manuscript.

5) In the discussion, I do not find any mention regarding the Little ICE Age (LIA). Indeed, from Fig. 2 we can observe an increase of Subalpine and Alpine species during the last 1000 years.

Our comment: We agree with reviewer 2 that this is a worthwhile point to discuss and thank him for pointing it out.

Action taken: We mention the Little Ice Age in the discussion (lines: 336-338).

Other points

L71: ... and forms elevational belts à weird sentence. I suggest to replace with "...determined by temperature, landforms and geomorphological processes along elevational belts". ...or similar sentence. Indeed, not only landforms influence vegetation but also the intensity of processes acting along slopes

Action taken: We changed the sentence as the reviewer suggested (lines: 75-77).

L74: not clear to what refers the term "present-day" in relation to alpine species. Considering the concept of "alpine life zone" alpine species can change the optimum elevation over time. So, I would remove "present-day"

Action taken: the word "present-day" was removed from the sentence.

L93: I suggest to replace immigration with "migration" since the word immigration implies only movement from the external regions to the Alpine area and not for instance internal movements from an alpine area to another one. I know that in zoology the term migration refers to seasonal movement of animals but this meaning do not apply for plants or humans.

Action taken: the word was replaced by "migration". (line: 94-96).

L163-185. Didn't' you perform any statistic test on the RDA analysis? E.g. variable selection, significant variables, variance etc. I think that to understand the most important variables is an added value

Our comment: we thank reviewer 2 for reporting this statistics on the RDA analysis. We performed the variable selection using ordistep and the AIC criterion, and we calculated Pillai's trace statistic to check the significance of the RDA components and the individual explanatory variables (Supplementary data S6).

Action taken: we added the statistics in the text and the Supplementary data S6.

Reviewer #3 (Remarks to the Author):

Summary

Garcés-Pastor et al. present a multi-proxy lake sediment record from the Swiss Alps that provides detailed insight into climate- and human-driven ecological change over the last

~12,500 years. The sedaDNA datasets (plant and mammal) are particularly rich and yield nuanced records of plant community composition and human activity that would not be possible with traditional analyses. The authors employ a couple of new approaches to interpret the sedaDNA data, including identifying local elevation band and pastoral/arable indicator taxa and introducing a useful metric, the “relative abundance index.” Their inferences about human impact are corroborated by traditional analyses of microcharcoal and coprophilous fungal spores. While the data presented are robust and yield interesting insights about human impacts in the Alps, the manuscript is currently a bit challenging to follow because of some missing context/results and a substantial amount of interpretation included prematurely in the results section.

Our comment: We thank the reviewer for these nice words and also for pointing out that it is sometimes a bit challenging to follow.

Action taken: We reorganized the results text to include a brief subsection for groups of proxies and excluded the interpretation included prematurely on the results.

Overall, I believe the paper will make a nice contribution to Nature Communications, but it would benefit from substantial revision of the text and minor updates to most of the figures. Below I summarize my major comments and line-by-line suggestions.

Major comments

- **Background and context of land-use:** Given that much of the focus of this paper is on the impact of human activities, more information on the recent/historical human occupation of this area (catchment and/or region of the Alps) is needed to place the proxy records in context. Does the timing and style of land use suggested by your dataset fit with what was already known about human activity here? Are there other proxy records from nearby sites or similar elevations in the Alps that corroborate this history? I recommend this be added either to the end of the introduction or a new first subsection within Results, with a brief comparison with other regional proxy records included in the Discussion.

Our comment: We thank the reviewer for reporting that, we added it.

Action taken: Information about the human occupation of the area was added to the Study site section. (lines: 457-465). We also added a sentence comparing our pollen results with the pollen regional trends (lines:179-183; 351-353).

- **Results/Discussion structure:** The manuscript is currently lacking a clear first-order description of much of the data (including the core lithostratigraphy/age model and the proxy datasets), and the results section contains quite a bit of interpretation (e.g., inferring the drivers of vegetation change) that would better fit into the discussion. As such, it's hard to follow the results section because it does not walk the reader through the results before diving into the more nuanced second-order takeaways. I suggest reorganizing the text to include a brief results subsection for each proxy (or group of proxies); then, the discussion could include a narrative with interpretations for each time period (as is currently the organization for the results section, but ideally more concise). Alternatively, this could

evolve into a "result and discussion" section, but I think that would be harder to follow without first being introduced to the general proxy results/patterns as part of a strictly "results" section.

Our comment: We agree with the reviewer that a better description of results might help the reader.

Action taken: A section for "Sediment core lithology and chronology", "Holocene temperature and precipitation development" and "Palynological and human indicator proxies" was added in the results (lines: 132-141; 177-192). We also deleted interpretations from the results.

- Interpretations about temperature driving plant richness change: The authors suggest that temperature increases during the early part of the record result in increased plant diversity and contrast this finding with Liu et al., 2021. However, this may be an overinterpretation of the RDA results; the time series of chironomid-inferred temperatures and sedaDNA-inferred plant richness (Fig. 2) shows that this relationship is nuanced and that the most prominent warming phase (~11 to 9 ka) corresponds to a steady decrease in richness, which supports the findings of Liu et al. I suggest that this interpretation be re-evaluated, at least with text changes but also perhaps with a more direct assessment of the correlation between temperature and richness.

Our comment: We thank the reviewer for pointing this out, we performed statistical correlation between richness, climatic and human drivers. Our results agree with Liu et al. and this was changed in the text.

Action taken: the new statistics were added to Figure 5 and text was changed according to the new statistical analyses (lines: 384-387).

Line-by-line comments

Line 52: "Biodiversity hotspot" has a specific definition based on number of endemics and threatened species; I'm not sure it's appropriate to apply it generally to "alpine areas." Consider rephrasing.

Action taken: "Biodiversity hotspot" was changed to "highly rich in species" (lines: 60-61).

Lines 60-61: Phrasing a bit awkward; consider changing to "more intensive agropastoralism" or something similar.

Action taken: the sentence was changed as suggested by the reviewer (lines: 65-68).

Line 77: I recommend splitting this into two paragraphs before "Plant remains..." with some sort of transition.

Action taken: the sentence was split into two as the reviewer suggested (lines: 82-84).

Lines 89-90: No need for quotes around "natural" and "human."

Action taken: quotes were deleted as suggested by the reviewer (line: 92-94).

Line 103: Change “composition” to “richness” (assuming that's what is meant).

Action taken: we changed “composition” to “richness” (lines: 117-118).

Lines 105+: Somewhere, it would be useful to include a bit of information about the historic/recent land use practices in this area to provide context.

Our comment: We thank the reviewer for pointing out about adding this information.

Action taken: A text was included on the Study site section (lines: 456-464).

Lines 114-126: This paragraph contains a lot of results for the introduction; it currently reads more like an extended abstract. I recommend revising to remove some of the results sentences and keep this more as a "roadmap" paragraph that introduces the study approach.

Our comment: We thank the reviewer for his comments about the introduction, so we can address it.

Action taken: We removed some of the result sentences from the introduction.

Line 128: It would be helpful to include a brief results section focused on the sediment core lithology and chronology, rather than jumping right to the proxy records. Some details about the lithologic changes and age model (including the notable increase in sedimentation rate around 5 ka) are important to understanding this record and should be mentioned at least briefly in the main text.

Our comment: We thank reviewer 3 for his helpful comment about providing a brief results section of all the proxies for a better structure of the manuscript.

Action taken: A paragraph for “Sediment core lithology and chronology” was added in the results (lines: 132-141).

Lines 158-160: Are the temperatures in parentheses means for the time period specified? If so, describe them as such. If not, it would probably be better to round to the nearest degree and give a range for that time period, as done on line 197.

Action taken: We added mean T in the text (new lines: 149-153).

Line 162: Before explaining the drivers of vegetation changes, there needs to be brief results sections describing all the proxy results (sedaDNA of plants and mammals, pollen, charcoal, coprophilous fungi). Determining the drivers of change is a second-order result that probably belongs in the discussion.

Our comment: We thank reviewer 3 for his comment about adding a brief results section

Action taken: Paragraphs for “Holocene temperature and precipitation development” and “Palynological and human indicator proxies” were added to describe the main proxies in the results (lines: 142-157; 177-192).

Line 169: I believe Fig. 3 has not yet been referenced (in part, because there is currently no results section describing much of the proxy data).

Our comment: We thank reviewer 3 for pointing this out

Action taken: A reference to Figure 3 was added in line 173, section “Improved identification of plants and ecological indicators from *sedaDNA*”.

Line 175-176: This type of interpretation belongs in the discussion, not the results section.

Action taken: The paragraph “Determining drivers of vegetation changes” was rewritten. (lines: 193-222).

Line 189-190: The base of the core definitely seems to be older than the end of the YD (11.7 ka) based on the age model. I would rephrase to say something like “likely originate from around the Younger Dryas event.”

Action taken: We added that sentence to the manuscript (lines: 228-230).

Line 195: Reference Fig. 2 when describing the *sedaDNA* results.

Action taken: Figure 2 was added in the sentence (lines: 230-232).

Line 197: There is no panel F or G on Fig. 2.

Action taken: The Figure was changed as suggested (line: 232).

Line 199-200: This is also interpretation that belongs in discussion. Also, temperature looks to be about the same (probably within error) in Zones 1 and 2 (as they are outlined in this figure).

Our comment: We thank the reviewer for pointing this out

Action taken: This sentence was rephrased so it is no longer an interpretation, as reviewer suggested (lines: 234-237).

Line 203: Figure reference should be SF8.

Action taken: Supplementary Figure SF7 was changed for SF8 (lines: 237-239).

Lines 203-206: More interpretation that belongs in discussion, here and in subsequent paragraphs.

Our comment: We thank the reviewer for pointing this out.

Action taken: Interpretation sentences were deleted from the results as suggested by the reviewer.

Lines 235-236: The fungal spores record shows similar flux values in the earliest part of the record, so I'm not sure I agree with this description/interpretation.

Our comment: The earliest part of the record presents a different erosion rate than the period we are referring to, as it can be seen in the age-depth model. Then, the beginning of the core could be influenced by a higher erosion that might have accumulated the fungal spores.

Action taken: A text was added to the sentence to point out that the sediment accumulation rate did not change in this period to show that this period is not comparable to the earliest part of the record.

Lines 244-245: To my eye, there is not much happening at 6.35 ka; it looks comparable to "background" levels of human indicators from first half of the record. There is no discernible change in microcharcoal. If you would like to point to an increase in grazing indicators, I recommend at least reducing the temporal precision ("around 6 ka", or simply remove that and leave only "during the latter half of the Neolithic (7.5-4.2 ka)") because this is not a very sharp transition. Also, move the Fig. 3 reference to the end of the sentence (with Fig. 2).

Our comment: We thank the reviewer for his comments.

Action taken: "fire" and "during the latter half of the Neolithic (7.5-4.2 ka)" were deleted from the sentence. Figure 3 was also added at the end of the sentence as suggested by the reviewer (lines: 274-277).

Line 254: Does the appearance of red deer suggest human impact? Please explain this connection more clearly.

Our comment: We consider that the appearance of red deer might be related with periods of less human impact.

Action taken: The sentence was changed to explain this point more clearly (lines: 281-286).

Line 320: Perhaps this should be "global or incomplete regional reference databases"?

Action taken: The sentence was changed to "global or incomplete regional reference databases" as suggested by the reviewer (lines: 346-348).

Line 344: For clarity, I would just change "p6 loop sequence" to something more general like "marker sequence" or "barcode sequence" for readers less familiar with these methods.

Action taken: We changed the "p6 loop" for "marker" (lines: 369-370).

Line 347-348: This sentence is a bit vague; perhaps give an example of how this indicator sequence approach could be applied?

Action taken: An example was added as suggested by the reviewer (lines: 374-377).

Lines 354-356: The highest richness of the early part of the record (~11-10 ka) precedes the major temperature rise and actually coincides with quite low temperatures. The RDA does point to climate being a driver of composition change during the first half of the record, but I'm not sure this statement about temperature and plant diversity specifically is supported by the data.

Our comment: We agree with the reviewer after the new Spearman correlation between richness and RDA axis.

Action taken: The sentence was rephrased to address the weak correlation between richness and climate drivers (lines: 384-387).

Lines 357-359: More accurately, Liu et al. reported a negative relationship between temperature and plant richness. And actually, I think there is similarity between the records when looking at the alpine/subalpine-dominated interval of the Sulsseewli record (up to ~9 or 9.5 ka). Both show relatively high richness corresponding to low temperatures, followed by a decrease in richness when temperature rises. It would be worth exploring this relationship more directly, and if the relationship changes once montane/colline vegetation is more abundant, that's an interesting result.

Our comment: We thank reviewer 3 for pointing this out, as a result, we explored this relationship more directly by doing a Spearman correlation of richness with the RDA 1A and RDA 2A of the species composition RDA analysis.

Action taken: The paragraph was changed according to the new statistical analysis (lines: 384)

Lines 360-361: I'm not convinced that different ID thresholds means that the two richness records aren't comparable, particularly when both studies are focusing on relative change through time (not comparing absolute richness).

Our comment: We agree with the reviewer with his comment.

Action taken: The sentence was deleted.

Line 363-365: Can you rule out that there is no transport of plant DNA from higher elevations down into the lake, given excellent preservation, high overall sed rates, and high clastic flux (potentially promoting more stable transport of mineral-bound DNA)? It would be prudent to point out this possibility at least. I believe that most modern sedDNA validation studies (e.g., Alsos et al., 2018) have been conducted in lower-relief settings, so it's hard to rule out enhanced DNA transport in this setting.

Our comment: We thank the reviewer for his comment. We could explain the presence of higher elevation taxa but wouldn't be able to explain the presence of lowland taxa without any type of human inference.

Action taken: We have rephrased the sentence to point out the erosion as the reviewer suggested (line: 390-392).

Lines 389-390: This topic sentence is very similar to that of the previous paragraph; modify to reflect that this paragraph is more about high levels of richness being associated with human activity.

Our comment: We thank the reviewer for reporting that, we added it

Action taken: We modified the sentence to better differentiate it from the one from the previous paragraph (line: 416-417).

Line 392: "Millennia" should be "millennium"

Action taken: We changed the word as suggested (line: 416-419).

Lines 399-400: It's unclear what impacts you're referring to about here. Reduction in richness? Also, if you are comparing with Liu et al specifically (as indicated in next sentence), say that instead of "studies elsewhere," which is vague.

Our comment: We thank the reviewer for reporting that, so we can address it.

Action taken: We changed the sentence to make it clear that we compare to Liu et al results (line: 424-427).

Line 402: It's a bit unusual to say "our findings stand with"; I recommend changing to "our findings support."

Our comment: We agree with the reviewer

Action taken: We changed the words according to the reviewer comments (line: 424-427).

Line 405: I suggest briefly explaining "intensive" vs "extensive" in this context, as your readership may not be familiar.

Action taken: We rephrased the sentence to better explain intensive vs extensive (lines: 430-432).

Lines 410-412: It's confusing how the observation from the previous sentence is "in line with" studies looking at the relationship between richness and temperature, given that the previous sentence is about human land-use, not climate change. Please clarify/revise.

Action taken: We deleted "this is in line with" from the sentence.

Line 417: The phrasing "the within dominant alpine families" is unclear; please rephrase.

Action taken: The sentence was changed to "within the most common alpine plant families" (lines: 439-442).

Lines 418+: The flow of logic through this paragraph is difficult to follow. The previous sentence is about more precise metrics for tracking/documenting alpine biodiversity change. The transition “Therefore” suggests the next idea stems from that, but it is instead about promoting traditional land-use practices to conserve biodiversity, which is quite different. The transition between these sentences should be changed, and perhaps the more prescriptive conclusions about maintaining traditional land-use should be in its own paragraph.

Our comment: We thank the reviewer for his concern about this paragraph.

Action taken: The last paragraph was rephrased to more prescriptive conclusions about maintaining traditional land-use (lines: 443-446).

Lines 423-424: It’s rather unusual to bring up new ideas (mammalian impacts) in the final sentence of the manuscript main text. Consider adding a final concluding/forward-looking sentence.

Action taken: The sentence about mammal impact was removed.

Lines 431-432: Are there any outlets, or is it a terminal lake?

Action taken: The information was added to the Study site (lines: 453-454).

Lines 432-434: It would be more relevant to provide local climate parameters if possible, rather than lowland values.

Action taken: Local climate parameters were provided (lines: 454-455).

Lines 445-446: It’s very difficult to discern these lithostratigraphic markers in Fig. SF1. Please increase the resolution of this figure and the text annotations. What is the difference between the red lines (correlation points) and blue lines (lithostratigraphic features)? More information is needed on how the cores were correlated.

Action taken: The text was provided in the Core sampling, lithology and chronology (lines: 465-482).

Lines 467-471: This currently reads as though the regression model/temperature calibration was newly generated for this study, which I don’t think is the case. Please rephrase slightly to make it clear you used a previously published and broadly applied transfer function.

Action taken: The paragraph was rephrased to make it clear (lines: 502-517).

Line 510: “Generic” is a bit confusing; perhaps “universal” is better.

Action taken: The word was changed (lines: 553-557).

Line 561: Are the common lab contaminants that you removed included in a table? It’s not clear if these are included in table ST6.

Action taken: Information about common lab contaminants was added to the sentence (lines: 599-604).

Lines 583+: This is a useful new metric that may be applied in future studies (I assume you hope it will be!). As such, it would be very useful to more clearly define RAI using an equation, as it's currently hard to follow this description. This should include the definition of weighted PCR replicates, rather than referring the reader to another study.

Our comment: We thank the reviewer for pointing this out, this has improved our manuscript.

Action taken: The equation and definition were added to the text (lines: 636-644).

Figures:

Fig. 1: Label Switzerland. Specify in the caption that the lake is circled in red.

Action taken: Switzerland was labeled in Figure 1.

Fig. 2: In the caption, clarify what smoothing function is used in panel A. For panel D, add a label next to "D" for symmetry with the others (perhaps "Climate proxies"?). Somehow (with color?) make it clear which y-axis corresponds to LOI vs. July temp. Add "(%)" for LOI. Add standard error for air T estimates.

Our comment: We thank the reviewer for pointing this out, this has improved the figure.

Action taken: The "(loess-smoothed lines, span = 0.1)" was added to the caption. The -eSEP, +SEP and precipitations were added to Figure 2.

Fig. 3: Figure caption needs a description of the top panel (sedaDNA results binned by elevation band indicators). Reorder figure caption information to reflect the order in the figure itself (e.g., microcharcoal, arable plant sedaDNA, etc). It would probably make more sense to switch the positions of coprophilous fungi and pastoral sedaDNA so that the mammal indicators (and plant sedaDNA datasets) are grouped.

Action taken: The figure caption was adapted and the coprophilous fungi and pastoral sedaDNA were switched.

Fig. 4: Please explain what the "Groups" are – less than that age? It would be clearer to list the age range (e.g., <3 ka, 3–6 ka, etc.) included for each category, and replace "Groups" with "Age group". Is Colline belt not on the plot because it is beyond the limits of this plot? If so, perhaps draw an arrow pointing toward where it would plot. If not, explain more clearly what you mean here.

Our comment: We thank the reviewer for pointing this out, so we can address it

Action taken: Figure 4 was changed according to the new statistical approach.

Table ST1. Please add a column with median calibrated age (or 95% range) so this information can more easily be compared with the age-depth model.

Action taken: A column with the “median calibrated age (cal BP)” was added in table ST1.

Fig. SF5: Define “LPAZ.”

Action taken: LPAZ was defined in Supplementary Figure SF5.

Table ST2: The “percentage” column appears empty – are values missing here? If so, please explain in the caption what this percentage refers to. If not, remove.

Our comment: We thank the reviewer for pointing this out, so we can address it

Action taken: The column was deleted from Supplementary table ST2.

Table ST5: I could not find a comparable table for arable indicators. Perhaps these could be incorporated into the same table, structured more like Table ST2.

Our comment: We thank the reviewer for pointing this out, so we can address it

Action taken: A column with the equivalent pollen taxa was added in Supplementary table ST5.

REVIEWER COMMENTS

Reviewer #1 (Remarks to the Author):

Thanks for your response and time.

Unfortunately, the DNA degradation problem, the statistical problems, and the speculation issues still persist as authors either did not address those or the new analyses and data confirmed my concerns. The persistence of those problems strongly impairs the reliability of results and the robustness of the manuscript.

Ignored, unresolved problems:

1. Your analysis (RDA) requires linear relationships among independent and identically distributed variables.

It is unhelpful to say “Our data are indeed linear, but we forgot to report that”. The data in Figure S8 and S11 are linear. Taxonomic richness data likewise their relationship with temperature and land-use intensity are NON-LINEAR. Thus, again, you violate the assumptions of RDA. This means that results are not reliable.

2. Once more, there is spatiotemporal autocorrelation in your data. To make this point even more clear: time series or spatiotemporal data like yours are autocorrelated, suffer autocorrelation. This is the case because there are spatial and temporal effects occurring in the intrinsic nature of the sedimentation and extraction/reconstruction processes. This means that your data are NOT independent. Indeed, two consecutive points in time/space are not independent being more likely to be similar to each other than two points that are further away in time/space. So, as already mentioned, you need to account for that, and there is a plethora of statistical tools to do so as highlighted previously (e.g., GLS, AR1 in mixed models).

3. Your variables are not identically distributed as explained below in relation to heterogeneity of variances.

As a result, the concomitant violation of these three assumptions strongly impairs the reliability of results. These concerns were not addressed in revision.

4. Still, authors introduced confusion around climate vs human-driven influence on plant diversity. The two factors do not exclude each other as a more benign climate during the Holocene optimum favoured an increase in human population as well as ecosystem productivity that ultimately lead to an increase in the intensity of human activities and pressures. Thus, the effect of climate is both direct and indirect, mediated by a third factor (productive environments and human density). Furthermore, as already stated, claims around human agropastoral activities during 6 ka, 3.5 ka and 1.5 ka increasing plant richness are not supported by data. Again, as one can clearly see in Fig. SF8, plant richness oscillates all the time

and, on average, does not increase from 11 ka to 4 ka or 2 ka. So, not only plant richness does not increase during the last 6 thousand or 2 thousand years, but even less human activities contributed to such (un)increase. Yet, as in their response letter and the manuscript, authors keep arguing not only that humans are the sole factor driving diversity but even that they increased it.

For discerning partially-associated factors and properly attributing to humans (or climate) a role in observed patterns, see e.g.

<https://www.nature.com/articles/421891a>

<https://www.nature.com/articles/nature03089>

<https://www.science.org/doi/10.1126/science.aba0690>

<https://link.springer.com/article/10.1007/s10584-013-0873-6>

5. I highlighted some concerns such as the use of the word “tradition”. In the rebuttal letter, the authors wrote that they deleted it, while actually they are still using it in the manuscript.”

New issues:

1. Thanks for including precipitation in the analysis. As I suspected, it does play a major role. Connected to my previous point, the new analyses reported in Fig. 4/ Fig. SF12 indicate that, in one case (panel A), precipitation is as much as important as sheep and cows along the first axis and much more important than all other variables along the second axis. In the second case (panel B), precipitation and temperature are by far the most important variables. Those results are not only not addressed by authors, but even contradict their own claims.

2. New Figure 2 and Figure 5:

Total DNA trend is problematic, highlighting an unbearable issue: DNA preservation in the core. Because DNA degrades with increasing time, samples will have different DNA degradation rates. Just because of this differential DNA preservation/degradation in the core and but not due to actual ecological processes, recent cores have higher total amount of DNA. I already highlighted this problem and the new results in Fig. SF10 and SF11 unfortunately confirm it. Indeed, such problem is clearly evidenced in your new Figure SF10, panels A-D. There, one can see that older sediments have a much lower quality of DNA sequences than newer sediments. Not only the average DNA quality of old sediments is poorer, but also the variance is much higher. This indicates that taxa over/under-representation due to differential DNA degradation/preservation is much problematic in older than younger sediments.

3. Nevertheless, authors still consider total DNA, which does not account for this issue, but, on the contrary, it exacerbates it. Total plant taxonomic richness from pollen data suffers the same problems. In addition to that, when smoothing the data, it appears clear that no actual increase trend occurs 6 ka as opposed to what is continuously claimed by authors.

4. Furthermore, these two new figures highlight that variance is not homogeneous, requiring statistical techniques accounting for heterogeneous variance components.

5. Finally, in new Fig. 5, both panels, the vast majority of points lie outside the confidence interval, highlighting very poor power of the model. Thus, the model they used is poor and should be replaced by GLS.

In conclusion, patterns reported in the new Figure 2, panel A and associated results in new Fig. 5 arise mainly from artifacts and not from an actual effect of climate or land use change.

Reviewer #2 (Remarks to the Author):

I found the revised version of the manuscript “High resolution ancient sedimentary DNA shows that regional alpine plant diversity is a result of human land use” really improved. Most of the reviewers’ comments have been addressed adequately. As already written for the previous revision, I found that the manuscript is quite interesting, well written and presented. However, since I’m also required to report about the novelty of the work, I think that the methodological approach is moderately novel since the first work on sedaDNA were presented about 8 year ago. The same applies to the main message of the manuscript that was already reported by previous works. Said that, the approach proposed by the authors have captured some original aspects (genetic database, integrated approach, high resolution, number of taxa) that perhaps can justify its publication in Nature Communications.

Specific points.

Result/Discussion. With regard my previous comment on refugia, I was aware that the data presented are not based on genetic evidences. However, I think that the investigation of your database, for what concerns the presence of relict plant species diversity and abundance, could disclose (or not) refugial character of the investigated site. Specifically, high-resolution studies (how this one) may help in defining migration dynamics of specific taxa, revealing extinctions, migrations and detecting possible new refuge areas. I think this aspect could make the manuscript more interesting and enhance the value of the data collected

L94. I think it is no necessary to put the name of the vegetation belt (Nival belt, Colline belt) in upper case

L295. Of species favoured by human disturbances such as *Alnus alnobetula*. I’m not so sure *Alnus alnobetula* is a good example. This species is especially favoured by snow avalanche and running water. I suggest to replace with another species

Fig 1D. Is it right that in southern alps vegetation belts reach lower elevation in the southern slope than in the northern one? Please, check.

Figure 4. “Coprophilous Fungi” in Fig. 3 is “Coprophilous fungi” (i.e. fungi lower case). Please be consistent.

Reviewer #3 (Remarks to the Author):

Garcés-Pastor et al., Nature Communications
Round 2 review

Summary:

The revised version of this manuscript is substantially improved, and I am satisfied with most the changes made to address my first-round comments. The structure and flow of the text are improved, and the revised statistical analyses and corresponding interpretations strengthen the manuscript. I will note that there is still some interpretation in the Results section, but I will defer to the editor and journal to determine whether that needs further attention. Below I provide a line-by-line summary of some remaining minor issues that should be addressed before publication.

Line-by-line comments:

Line 70-71: The phrase "...plant richness to levels that characterize diversity" is a bit confusing and vague; consider revising. (Do you mean "characterize the current high diversity of this region" or something?)

Lines 71-72: Consider rephrasing this slightly so that it is not quite so prescriptive; something like: "A continuation of agropastoral activities is most likely to preserve the unique plant diversity..." As it is currently written, it reads more like a perspective piece (and I somewhat agree with Reviewer 1 that this is a potentially controversial perspective).

Line 92: The phrase "diversity processes" is a bit vague and unconventional; consider rephrasing.

Lines 130-132: Here again, I suggest changing from prescriptive to descriptive: "The maintenance of traditional land use practices would help to preserve current high plant diversity of alpine ecosystems in the face of ongoing climate warming."

Line 140: The median calibrated age of the lowermost sample is more like 12.4 ka, not 11 ka. See comment for Table ST1 below for more detail. You could say that more generally, the age model has large uncertainty below ~11 ka.

Line 186+: It's a bit odd that the pollen and mammalian sedaDNA results are lumped into one paragraph. I recommend reorganizing such that all sedaDNA results are together, and pollen is in a separate paragraph; or at least, split this paragraph into two at around line 186.

Line 232-233: Please add a reference here to support the described climate patterns through the Younger Dryas.

Line 237: I don't think that the stratigraphic zones have been explained yet at this point; please clarify in the text how the zones were determined.

Line 330-331: I recommend adding a few words here to explicitly state how high erosion could favor transport of high-elevation plant DNA to the lake.

Line 372: Add the general location of the Liu et al. study here, as it's the first mention of this paper.

Line 611-613: Please specify in Table ST6 all the taxa that were retained for full transparency. 619-620: It would be helpful to include the MTQ/MAQ scores for the negatives in table ST6.

Line 642-645: Clarify if you removed or retained these sporadic detections; one of the cow occurrences mentioned is visible in Fig. 3, but not the others.

Fig. 1: As previously requested, please specify in the caption that the lake site is circled in red.

Fig. 2: Define eSEP in the caption and make sure the figure version matches the caption (SEP vs eSEP; I believe the plus and minus are swapped in the figure as well).

Fig. SF2: It is still difficult to discern the lithostratigraphic markers in this figure, and the imagery is not sufficiently high resolution to identify the events used for correlation. It is thus impossible to evaluate the quality of the core alignment. I recommend reformatting this figure to be split into ~three columns so that the images can be larger and higher quality.

Table ST1: The calibrated age column appears to be incorrect. The lowermost dated sample has a ^{14}C age of 10590 ± 250 , which yields a 95% calibrated range of 13075-11730 cal yr BP (median of ~12.4 ka). The reported median of 11814 is perhaps the age model estimate for this depth, rather than the actual calibrated age of that sample. Please check to make sure that the ages in this column are the calibrated ages of the samples listed, not the age model interpolation.

Response to reviewers

Dear reviewers,

We thank you very much for your nice comments, which have contributed to considerably improve the quality of the manuscript.

We have performed additional statistical analyses suggested by the reviewers and our new more robust statistical results do not influence our original conclusions.

The authors have addressed all the reviewers' suggestions, which have improved the structure and content of the manuscript, and we respond to them point-by-point below.

Yours,

Sandra Garcés-Pastor and co-authors

Updated figures and tables:

Revised manuscript

Figure: 2, 3, 4, 5

Supplementary Figure: SF1, SF8, SF10, SF11, SF12, SF13

Supplementary Table: ST1, ST6, St8, ST9, ST10,

Supplementary dataset: S1, S3, S3.1, S3.2, S4, S5, S6, S7

Reviewer #1 (Remarks to the Author):

Thanks for your response and time.

Unfortunately, the DNA degradation problem, the statistical problems, and the speculation issues still persist as authors either did not address those or the new analyses and data confirmed my concerns. The persistence of those problems strongly impairs the reliability of results and the robustness of the manuscript.

Ignored, unresolved problems:

1. Your analysis (RDA) requires linear relationships among independent and identically distributed variables.

It is unhelpful to say "Our data are indeed linear, but we forgot to report that". The data in Figure S8 and S11 are linear. Taxonomic richness data likewise their relationship with temperature and land-use intensity are NON-LINEAR. Thus, again, you violate the assumptions of RDA. This means that results are not reliable.

Our comment: Thank you for your continued scrutiny of this issue. In our new analyses, we separately analysed taxonomic composition and richness data. We followed standard protocols (Ter Braak and Prentice 1988) while selecting ordination techniques to analyse compositional data.

We understand that there are no linear processes in nature but RDA analysis has been extensively used in palaeoecological studies as a robust way to describe and visualise the patterns of the data. However, we have now performed additional analyses that further support our conclusions.

We have now also used generalised additive models (GAMs) to explore the relationship between taxonomic richness and different explanatory variables. GAMs capture the non-linear covariate effects which are common in palaeoecological records (Simpson 2018).

Action taken: The composition data was used to estimate the length of the environmental gradient using detrended correspondence analysis. The gradient length was used as the basis for deciding linear or unimodal direct gradient analysis. Given that the gradient length of the first DCA axis was 1.9 SD, we used RDA as recommended by Ter Braak and Prentice (1988).

In addition, we performed a piecewise redundancy analysis (Vieira et al 2019) in order to check if we could retrieve any improved ecological interpretation from our data. Results of this analysis do not differ from those of the original RDA analysis (see figure below, full RDA in the left and piecewise RDA in the right), so they have been left out of the manuscript to avoid the introduction of additional complications for the sake of simplicity.

Finally, we built GAM models considering taxonomic richness as the response, and both the climatic variables and proportion of domestic animals as the explanatory variables. Given that our response is count of taxa, we used a poisson distribution and log link in the GAM models. These results have been included in the manuscript.

- Simpson, G. L. Modelling palaeoecological time series using generalised additive models. *Front. Ecol. Evol.* 6, (2018).
- Ter Braak, C. J. F. & Prentice, I. C. A theory of gradient analysis. in *Advances in ecological research volume 18* vol. 18 271–317 (Elsevier, 1988).
- Vieira, D. C., Brustolin, M. C., Ferreira, F. C., & Fonseca, G. (2019). segRDA: an R package for performing piecewise redundancy analysis. *Methods in Ecology and Evolution*, 10(12), 2189-2194.

2. Once more, there is spatiotemporal autocorrelation in your data. To make this point even more clear: time series or spatiotemporal data like yours are autocorrelated, suffer autocorrelation. This is the case because there are spatial and temporal effects occurring in the intrinsic nature of the sedimentation and extraction/reconstruction processes. This means that your data are NOT independent. Indeed, two consecutive points in time/space are not independent being more likely to be similar to each other than two points that are further away in time/space. So, as already mentioned, you need to account for that, and there is a plethora of statistical tools to do so as highlighted previously (e.g., gls, AR1 in mixed models).

Our comment: Thank you for pointing out this issue. We explored the extent of auto-correlation in the residuals of our model using the Durbin-Watson test. We found an indication of negative correlation between residuals at lag-1. However, the autocorrelation was not very strong (non-significant Durbin-Watson statistics) in our model. Given that our model will not be used for further predictive modelling, we consider that the non-significant autocorrelation of residuals can remain in the model.

Action taken: We interrogated our GAM models for the presence of temporal auto-correlation. Given that there was no evidence of the presence of strong autocorrelation in the model, we accepted the model as is. We describe these findings in the Methods (lines: 630-644).

3. Your variables are not identically distributed as explained below in relation to heterogeneity of variances.

As a result, the concomitant violation of these three assumptions strongly impairs the reliability of results. These concerns were not addressed in revision.

Our comment: We are aware that palaeoecological records seldomly fulfil the requirement for parametric statistical tests. However, there are other approaches that can be used for palaeoecological data such as ours. We consider GAM as

one of the approaches that can be used to analyse non-normally distributed data where non-linear effects of covariates are also present.

Action taken: We have used generalised additive models (GAMs) to explore the relationship between taxonomic richness and different explanatory variables. GAMs capture the non-linear covariate effects which are common in palaeoecological records (Simpson 2018).

- Simpson, G. L. Modelling palaeoecological time series using generalised additive models. *Front. Ecol. Evol.* 6, (2018).

4. Still, authors introduced confusion around climate vs human-driven influence on plant diversity. The two factors do not exclude each other as a more benign climate during the Holocene optimum favoured an increase in human population as well as ecosystem productivity that ultimately lead to an increase in the intensity of human activities and pressures. Thus, the effect of climate is both direct and indirect, mediated by a third factor (productive environments and human density). Furthermore, as already stated, claims around human agropastoral activities during 6 ka, 3.5 ka and 1.5 ka increasing plant richness are not supported by data. Again, as one can clearly see in Fig. SF8, plant richness oscillates all the time and, on average, does not increase from 11 ka to 4 ka or 2 ka. So, not only plant richness does not increase during the last 6 thousand or 2 thousand years, but even less human activities contributed to such (un)increase. Yet, as in their response letter and the manuscript, authors keep arguing not only that humans are the sole factor driving diversity but even that they increased it.

For discerning partially-associated factors and properly attributing to humans (or climate) a role in observed patterns, see e.g.

<https://www.nature.com/articles/421891a>

<https://www.nature.com/articles/nature03089>

<https://www.science.org/doi/10.1126/science.aba0690>

<https://link.springer.com/article/10.1007/s10584-013-0873-6>

Our comment: We thank the reviewer for their comments on the plant richness trends in Figure SF8. As we describe below, we now have an additional conservative filtering strategy to scrutinise these richness patterns. We find that the 6 ka increase was likely an artefact of DNA data quality, whereas the 2 ka increase persists and is now clearer.

Action taken: We have updated Figure SF8 and the text on richness trends accordingly (lines: 244, 259, 267, 392).

5. I highlighted some concerns such as the use of the word “tradition”. In the rebuttal letter, the authors wrote that they deleted it, while actually they are still using it in the manuscript.”

Our comment: We thank the reviewer for pointing this out and apologise for this oversight.

Action taken: We have changed the word “traditional” in the text.

New issues:

1. Thanks for including precipitation in the analysis. As I suspected, it does play a major role. Connected to my previous point, the new analyses reported in Fig. 4/ Fig. SF12 indicate that, in one case (panel A), precipitation is as much as important as sheep and cows along the first axis and much more important than all other variables along the second axis. In the second case (panel B), precipitation and temperature are by far the most important variables. Those results are not only not addressed by authors, but even contradict their own claims.

Our comment: With the new analyses sheep becomes the most important variable for RDA 1A in species composition (Figure 4A), followed by precipitation, charcoal, and cow. With the strict quality filter, cows become the most important and the influence of precipitation disappears. This points to the same human influence in RDA 1A as we reported, indicating the robustness of our results. For the RDA 2A, LOI and wild animals become important, while precipitation and temperature lead with the strict filtering. The signal of RDA 2A in both approaches is led by climate.

For the altitudinal vegetation belts (Figure 4B), precipitation is the most important variable of RDA 1B, followed by LOI, while cow and sheep lead RDA 2B. Their signal is the same for both quality filter analysis.

Action taken: The results of the new RDA analysis according to filtering criteria were updated in the text. (lines: 193-216).

2. New Figure 2 and Figure 5:

Total DNA trend is problematic, highlighting an unbearable issue: DNA preservation in the core. Because DNA degrades with increasing time, samples will have different DNA degradation rates. Just because of this differential DNA preservation/degradation in the core and but not due to actual ecological processes, recent cores have higher total amount of DNA. I already highlighted this problem and the new results in Fig. SF10 and SF11 unfortunately confirm it. Indeed, such problem is clearly evidenced in your new Figure SF10, panels A-D. There, one can see that older sediments have a much lower quality of DNA sequences than newer sediments. Not only the average DNA quality of old sediments is poorer, but also the variance is much higher. This indicates that taxa over/under-representation due to differential DNA degradation/preservation is much problematic in older than younger sediments.

Our comment: We thank the reviewer for further highlighting the DNA data quality issues. We note that we already removed the worst performing samples (n=6) that were mostly concentrated in the lower-most/oldest portions of the core. These six were not previously included in any downstream analyses.

However, there is still variance in the quality of the remaining retained samples that resulted in significant associations between sample age and measures of DNA quality (including both MTQ and MAQ scores). To test the robustness of our results, we have now divided our retained samples into two groups: those that pass a 'relaxed' filter (as previous, $MTQ \geq 0.45$, $MAQ \geq 0.175$) and those that pass a 'strict' filter (requiring both MTQ and MAQ scores ≥ 0.9). This latter filter removed both an additional 18 samples and the significant trends observed between the all data quality metrics and sample age, with the exception of mean wtRep which undergoes a shift in samples beginning c. 3.0-2.0 ka (Supplementary dataset S7; Figure SF8). We emphasise that the mean wtRep trend is otherwise flat between 12.0 and 3.0 ka, which does not support the hypothesis of an age-induced decrease in DNA data quality in the strict dataset.

Action taken: We have updated Supplementary Figures S10 and S11, Data S3 and the text: "Given the significant associations between sample age and four of the six measures of plant *sedaDNA* data quality, including both MTQ and MAQ scores (Supplementary dataset S7, Supplementary Figure SF10; SF11), we additionally divided our retained samples into two groups: those that passed a "relaxed" filter (as previous, $MTQ \geq 0.45$, $MAQ \geq 0.175$; $n=74$ samples) and those that passed a "strict" filter (requiring both MTQ and MAQ scores ≥ 0.9 ; $n=56$ samples). This latter filter removed both an additional 18 samples and the significant trends observed between the all data quality metrics and sample age, with the exception of mean wtRep which undergoes a shift in samples beginning c. 3.0-2.0 ka (Supplementary dataset S7; Supplementary Figure SF10). We emphasise that the mean wtRep trend is otherwise flat between 12.0 and 3.0 ka, which does not support the hypothesis of an age-induced decrease in plant *sedaDNA* data quality in the strict dataset. We note that both filtering strategies gave the same broad RDA results and so we report the "relaxed" results here and present results from the "strict" analysis in Supplementary dataset S6. However, for analyses of taxonomic richness (GAMs, Total DNA trend in Figure 2), we used the "strict" filtered data set due to the sensitivity to outliers in these analyses." (lines: 630-644).

3. Nevertheless, authors still consider total DNA, which does not account for this issue, but, on the contrary, it exacerbates it. Total plant taxonomic richness from pollen data suffers the same problems. In addition to that, when smoothing the data, it appears clear that no actual increase trend occurs 6 ka as opposed to what is continuously claimed by authors.

Our comment: Given our new filtering strategy, resulting in 'relaxed' and 'strict' datasets, we now use only use the strict dataset for analyses of *sedaDNA*-derived taxonomic richness (=Total DNA in Figure 2; the new GAM analyses, Figure 5). With regard to the trend reported in Figure 2, the original dip in richness between ~10 and 6 ka disappears using this strict filtering, in agreement with the suspicions of the reviewer. However, the increase in richness ~2 ka is unaffected by filtering strategy.

For the RDA analysis, we compared both the strict and relaxed data sets, which showed that the results were in broad agreement. We clearly state in the Results section (lines 194-217) where these two analyses diverge. Overall, both species

composition RDA analyses show that RDA1 was related to human drivers, whereas RDA2 was related to climate drivers.

Action taken: We have also updated Figures 4 and 5, Supplementary table S6.

4. Furthermore, these two new figures highlight that variance is not homogeneous, requiring statistical techniques accounting for heterogeneous variance components.

Our comment: Heterogeneous variance in palaeoecological data is quite common due to unequal number of years represented by the same thickness of the sample analysed due to differential sedimentation rates. However, models such as GAMs should be able to handle such an issue.

Action taken: We reanalysed richness data using GAM models and the figure 5 is updated. (lines 897-901).

5. Finally, in new Fig. 5, both panels, the vast majority of points lie outside the confidence interval, highlighting very poor power of the model. Thus, the model they used is poor and should be replaced by gls. In conclusion, patterns reported in the new Figure 2, panel A and associated results in new. Fig. 5 arise mainly from artifacts and not from an actual effect of climate or land use change.

Our comment: We now use generalised additive models (GAMs) as this approach handles our data better than the generalised least squares (gls). However, we appreciate the reviewer's suggestion, which was a better alternative than the linear regression that we used in the previous revision.

Action taken: We reanalysed richness data using GAM models and Figure 5 is updated (lines 897-901).

Reviewer #2 (Remarks to the Author):

I found the revised version of the manuscript “High resolution ancient sedimentary DNA shows that regional alpine plant diversity is a result of human land use” really improved. Most of the reviewers’ comments have been addressed adequately. As already written for the previous revision, I found that the manuscript is quite interesting, well written and presented. However, since I’m also required to report about the novelty of the work, I think that the methodological approach is moderately novel since the first work on sedaDNA were presented about 8 year ago. The same applies to the main message of the manuscript that was already reported by previous works. Said that, the approach proposed by the authors have captured some original aspects (genetic database, integrated approach, high resolution, number of taxa) that perhaps can justify its publication in Nature Communications.

Specific points.

Result/Discussion. With regard my previous comment on refugia, I was aware that the data presented are not based on genetic evidences. However, I think that the investigation of your database, for what concerns the presence of relict plant species diversity and abundance, could disclose (or not) refugial character of the investigated site. Specifically, high-resolution studies (how this one) may help in defining migration dynamics of specific taxa, revealing extinctions, migrations and detecting possible new refuge areas. I think this aspect could make the manuscript more interesting and enhance the value of the data collected

Our comment: We thank the reviewer for his comment and agree that exploring the classical biogeographic theories like vegetation refugia are very interesting. Unfortunately, a single lake study is not suitable to properly explore such idea. As more high-resolution studies become available, it will be possible to study vegetation refugia in depth.

Action taken: we added a sentence highlighting the potential of this high-resolution studies to define migration dynamics and better understand refuge areas. “As more high-resolution data like those generated here become available, our understanding of the migration dynamics of specific taxa and rates of extinction will increase and it may be possible to detect new refugial areas” (lines 378-380).

L94. I think it is no necessary to put the name of the vegetation belt (Nival belt, Colline belt) in upper case

Our comment: We consider that names should be capitalised because we are talking about that vegetation community in particular.

L295. Of species favoured by human disturbances such as *Alnus alnobetula*. I’m not so sure *Alnus alnobetula* is a good example. This species is especially favoured by snow avalanche and running water. I suggest to replace with another species

Our comment: We thank the reviewer for his concern about *Alnus alnobetula*. Today the species is favoured by both land use changes (in an intermediate succession phase between pastures and afforestation) and natural disturbance (avalanches, landslides, but not much running waters). During the Holocene, *Alnus alnobetula* shows close numeric connections with human impact in the Western Alps (towards the east including the Engadine, Inn-Valley). This connection became somewhat less pronounced during the past 2000 years, when the use of fire was reduced and subalpine meadows expanded at the expense of *Alnus alnobetula*. The sentence is correct as such from a Holocene perspective, where all mass expansions of *Alnus alnobetula* were connected due to excessive disturbance, especially during the Neolithic, Bronze Age and Iron Age. Thus it depends on the perspective, today or Holocene? Even for today the prevalence of *Alnus alnobetula* stands is mostly connected to the conversion of pasture into shrublands as a consequence of decreasing grazing by domesticated animals.

Action taken: we added another species (*Plantago lanceolata*) to support the rise of human related taxa. (lines: 296-297).

Fig 1D. Is it right that in southern alps vegetation belts reach lower elevation in the southern slope than in the northern one? Please, check.

Our comment: We understand the reviewer's remark since this is indeed apparently counterintuitive. However, the phenomenon has been well known for a long time, and especially with observations about the elevation of the treeline and the snowline during the nineteenth century. Basically, it is the result of the "mass elevation effect" ("Massenerhebungseffekt" in German) in mountain ranges, namely there is a greater mass of mountains in the centre of the range that extends at a higher mean elevation above the lowlands than the mean elevation of the front ranges. With this, the centre of the mountain range is also drier in terms of quantity of precipitation and air humidity, allowing a stronger irradiation, hence a greater warming of all surfaces that induces the isotherms to move upslope, contrary to the front range that is cooler due to condensation and high precipitation because air masses are forced going upslope (see e.g. Körner, C., 2012, Alpine treelines, Springer, Basel). Consequently, the greater the mass and the elevation in, and the extent of the centre of a mountain range, the higher the vegetation belts move up.

Action taken: we have provided references that provide the different elevations in the southern and northern slope of the Alps.

Fig. 3.3. Schematic representation of the 'massenerhebungseffekt' in large mountain systems

Fig. 3.4. The latitudinal variation of treeline and snowline modelled by climatic drivers (Körner 2007a; see also Chap. 5). Note the parallel trend in the biological boundary (treeline) with the purely physics-driven snowline

Figures extracted from Körner, 2012, p. 24-25

- Körner, C. (2012). Alpine treelines: functional ecology of the global high elevation tree limits. Springer Science & Business Media.

Figure 4. “Coprophilous Fungi” in Fig. 3 is “Coprophilous fungi” (i.e. fungi lower case). Please be consistent.

Our comment: We thank reviewer 2 for pointing out this inconsistency.

Action taken: Figure 4 and SF12 were changed according to the reviewer comments.

Reviewer #3 (Remarks to the Author):

Garcés-Pastor et al., Nature Communications
Round 2 review

Summary:

The revised version of this manuscript is substantially improved, and I am satisfied with most the changes made to address my first-round comments. The structure and flow of the text are improved, and the revised statistical analyses and corresponding interpretations strengthen the manuscript. I will note that there is still some interpretation in the Results section, but I will defer to the editor and journal to determine whether that needs further attention. Below I provide a line-by-line summary of some remaining minor issues that should be addressed before publication.

Line-by-line comments:

Line 70-71: The phrase “...plant richness to levels that characterize diversity” is a bit confusing and vague; consider revising. (Do you mean “characterize the current high diversity of this region” or something?)

Our comment: We thank the reviewer for the comment and have corrected the sentence according to their suggestion.

Action taken: the sentence was changed as suggested by the reviewer (line: 70).

Lines 71-72: Consider rephrasing this slightly so that it is not quite so prescriptive; something like: "A continuation of agropastoral activities is most likely to preserve the unique plant diversity...." As it is currently written, it reads more like a perspective piece (and I somewhat agree with Reviewer 1 that this is a potentially controversial perspective).

Our comment: We agree with the reviewer that this is a potentially controversial perspective.

Action taken: the sentence was changed accordingly (lines 70-71).

Line 92: The phrase "diversity processes" is a bit vague and unconventional; consider rephrasing.

Our comment: We thank the reviewer for pointing this out

Action taken: "diversity processes" was changed to "diversity changes" (lines 90-91).

Lines 130-132: Here again, I suggest changing from prescriptive to descriptive: "The maintenance of traditional land use practices would help to preserve current high plant diversity of alpine ecosystems in the face of ongoing climate warming."

Our comment: We thank the reviewer for their suggestion of a descriptive approach.

Action taken: The sentence was changed as suggested by the reviewer (lines 128-130).

Line 140: The median calibrated age of the lowermost sample is more like 12.4 ka, not 11 ka. See comment for Table ST1 below for more detail. You could say that more generally, the age model has large uncertainty below ~11 ka.

Our comment: We thank the reviewer for pointing this out.

Action taken: the sentence was changed as commented by the reviewer (line: 138).

Line 186+: It's a bit odd that the pollen and mammalian sedaDNA results are lumped into one paragraph. I recommend reorganizing such that all sedaDNA results are together, and pollen is in a separate paragraph; or at least, split this paragraph into two at around line 186.

Our comment: We thank the reviewer for pointing this out.

Action taken: We have split this paragraph into two (lines: 183-192).

Line 232-233: Please add a reference here to support the described climate patterns through the Younger Dryas.

Our comment: Thanks for the suggestion.

Action taken: The Baroni et al 2021 citation was inserted in the text (lines: 235-237).

- Baroni, C., Gennaro, S., Salvatore, M. C., Ivy-Ochs, S., Christl, M., Cerrato, R., & Orombelli, G. (2021). Last Lateglacial glacier advance in the Gran Paradiso Group reveals relatively drier climatic conditions established in the Western Alps since at least the Younger Dryas. *Quaternary Science Reviews*, 255, 106815.

Line 237: I don't think that the stratigraphic zones have been explained yet at this point; please clarify in the text how the zones were determined.

Our comment: We thank the reviewer for pointing this out.

Action taken: The text was included to clarify how the zones were determined "Seven vegetation zones were obtained from the constrained cluster analysis (CONISS) of *sedaDNA*". (lines: 230-231).

Line 330-331: I recommend adding a few words here to explicitly state how high erosion could favor transport of high-elevation plant DNA to the lake.

Our comment: We thank the reviewer for the suggestion to make the text more explicit.

Action taken: An explicit sentence was added to the text "The contribution from higher elevation taxa could have been favoured by a reduction in competition from shrub and tree species in open meadows, together with greater erosion due to deforestation that might have favoured transport of high-elevation plant *sedaDNA* to the lake." (lines: 330-332).

Line 372: Add the general location of the Liu et al. study here, as it's the first mention of this paper.

Our comment: We thank the reviewer for pointing this out

Action taken: "in the Tibetan Plateau" was added to the sentence. (lines: 372-373).

Line 611-613: Please specify in Table ST6 all the taxa that were retained for full transparency.

Our comment: We thank the reviewer for pointing this out

Action taken: A column of comments was added to Table ST6 to say that Picea abies and Pinus sequences were retained in the final dataset.

619-620: It would be helpful to include the MTQ/MAQ scores for the negatives in table ST6.

Our comment: Thank you for pointing out this omission.

Action taken: We have added the MTQ and MAQ scores for the controls to Supplementary Dataset S3 and have now signposted to this Dataset.

Line 642-645: Clarify if you removed or retained these sporadic detections; one of the cow occurrences mentioned is visible in Fig. 3, but not the others.

Our comment: We thank the reviewer for catching this. The sporadic detections were retained. However, two sporadic detections of cow are not visible in Figure 3: (1) the detection at 12.7 ka was beyond the confident age-depth model and so is not shown in Figure 3, and (2) the detection at 7.4 ka was based on very low read abundance and so is not visible in Figure 3 (which uses the RAI values). However, all detections are clearly visible in Supplementary Figure SF9.

Action taken: We have clarified the Figure 3 legend and signposted to Supplementary Figure SF9 in both the Figure 3 legend and the sentence in the methods section that the Reviewer highlights: The Figure 3 legend now states: "Note that the sporadic detection of Cow at ~12.7 ka is not plotted due to being beyond the confident age-depth model. Additional detections, including cow at 7.4 ka, are not visible due to low read counts but are visible in Supplementary Figure SF9.", and lines 883-885 state: "Although we retained detections from all samples, we interpret sporadic occurrences of these taxa, defined as single, isolated PCR replicate detections, as likely deriving from contamination"

Note that we have also corrected the ages of these samples with sporadic detections, which were out by 0.2 ka.

Fig. 1: As previously requested, please specify in the caption that the lake site is circled in red.

Our comment: Apologies for this omission.

Action taken: we specified in the caption that the lake site is circled in red (line: 850).

Fig. 2: Define eSEP in the caption and make sure the figure version matches the caption (SEP vs eSEP; I believe the plus and minus are swapped in the figure as well).

Our comment: We thank the reviewer for pointing this out.

Action taken: We changed Figure 2 according to the comments.

Fig. SF2: It is still difficult to discern the lithostratigraphic markers in this figure, and the imagery is not sufficiently high resolution to identify the events used for correlation. It is thus impossible to evaluate the quality of the core alignment. I recommend reformatting this figure to be split into ~three columns so that the images can be larger and higher quality.

Our comment: We thank the reviewer for the suggestion, which we think may be referring to Fig SF1.

Action taken: We modified Fig SF1 to improve clarity of the core alignment.

Table ST1: The calibrated age column appears to be incorrect. The lowermost dated sample has a ^{14}C age of 10590 ± 250 , which yields a 95% calibrated range of 13075-11730 cal yr BP (median of ~12.4 ka). The reported median of 11814 is perhaps the age model estimate for this depth, rather than the actual calibrated age of that sample. Please check to make sure that the ages in this column are the calibrated ages of the samples listed, not the age model interpolation.

Our comment: We thank the reviewer for the comment.

Action taken: We updated the values and changed the former column to “Median calibrated age” of the macrofossil radiocarbon date. Calculated with OxCal 4.4.

REVIEWER COMMENTS

Reviewer #1 (Remarks to the Author):

Thanks for taking the review seriously and addressing most concerns. This is a much improved and robust version. As anticipated, new analyses show that climate, and in particular precipitation, plays a major role in explaining biodiversity change and patterns. Furthermore, now it is clear that sedaDNA quality is a major shortcoming, but authors now did a good job in using higher quality data.

Unfortunately, with the GAM model we only know that precipitation and herbivores predict species richness (as indicated in table ST10), but we do not know HOW. That is, we cannot understand whether species richness increases or decreases with precipitation and herbivores, or whether there is a quadratic or exponential relationship among them. This is a crucial point. However, such key information is currently missing. Consequently, all statements around 'species richness increase with herbivores' remain unsupported by results. As such, those claims around livestock grazing lay in the domain of speculations. As already recommended, a gis-type of model would help you in answering such question and solving the problems highlighted previously as well as below.

A second unresolved problem is still associated to inferring quantitative data from sedaDNA. First, there are strong limitations associated with using reads frequency as proxy for species richness and abundance (see e.g. <https://onlinelibrary.wiley.com/doi/10.1111/mec.14920> and especially the book by Taberlet on eDNA). Second, erosion brings more sediment DNA into the lake, and erosion increases with deforestation and livestock grazing. So, higher estimated taxonomic richness may result from higher erosion and sediment DNA rather than from an actual increase in biodiversity. Unfortunately, all together, these shortcomings impair the reliability of results and solidity of associated claims.

To overcome such issue, as already mentioned several times, authors can use attribution models, identifying the relative contribution of different factors (climate, surface, ecological, anthropogenic). Those models are commonly done in climate science (see my previous review) and would make results more reliable.

Furthermore, authors keep ignoring the evidence that climate explains vegetation patterns. As a matter of facts, when excluding poor DNA data, the association between human land use proxies and vegetation disappears (see lines 217-219 and lines 222-224). Instead, climate always predict vegetation both in poor- and better-quality data as well as both in rda and gam models (see lines 227-229).

Please adjust the title accordingly by including climate too. Furthermore, given the descriptive, correlative nature of your data, claims about causal relationships should be avoided. This holds true starting from the title, where "is a result of" should be replaced by "is associated with". The new title shall read like "High resolution ancient sedimentary DNA suggests that regional alpine plant diversity is associated with human land use and climate".

Specific edits:

Lines 66-67 and elsewhere: according to the new analyses, there are no associations between climate and vegetation patterns as of 6 k BP. Please pay attention to stick to your actual results.

Lines 69: Where are the data showing a “reduced forest cover”? There is no evidence in support of a “reduced forest cover”. Delete.

Lines 70: “thereby increasing” – once more, please bear in mind that causal processes cannot be discerned with the current analyses. As already mentioned in my previous review, if authors aim at causal inference they should follow a. This is commonly done in climate science where the effects of anthropogenic impact and natural variation are parsed out and attributed to climate change. But such procedure is not unfortunately done in this manuscript. Given the current type of simplistic correlative models (RDA, GAM etc), it is not possible to tease apart the relative contribution of different predictors. Please adjust your language and terminology accordingly and avoid conveying misleading messages.

Lines 69–72: These sentences bear many different problems at once. In addition to those just explained above, authors keep pushing their ideological agenda toward deforestation and livestock grazing, ignoring the vast and deep ecosystem damages that those two activities bring. There is a vast literature highlighting the impact of livestock grazing across ecological scales that reduces biodiversity and multiple ecosystem functions. See e.g.:

-

https://onlinelibrary.wiley.com/doi/full/10.1111/ele.13527?casa_token=Tj45v4QN748AAAAA%3A3D7DpXq_e997jh3lzZPh7jJNKQIdIH3RKv67nNeDEMmBr2DxGC1-aJHgVcPChdHeYWzu_XkB9KnpW4

- <https://esajournals.onlinelibrary.wiley.com/doi/full/10.1890/ES14-00316.1>

- https://esajournals.onlinelibrary.wiley.com/doi/full/10.1890/15-1234?casa_token=cKyPr3jOcQEAAAAA%3AsJ3BUzyGbl1Vq-j0BPhG7RDx4HnZVPPlg-_LKNLOuXKR_jctDjXOyqoe8ZeT5x5ENs-7iMpOuXt--1g

-

https://onlinelibrary.wiley.com/doi/full/10.1111/rec.12676?casa_token=as9BTd3FWbAAAAAA%3ADXsOGK1rKbzuBYmUvmc5p2yhmJK490uPzgf1pVgd5-_VH-eyS-tisO_-1Sm-rZhVEXzQuphOP6Tq19s

-

https://onlinelibrary.wiley.com/doi/full/10.1002/ldr.2668?casa_token=q9IYAf11Zi8AAAAA%3ABmUcnkNIMoXNozW9no35Wchgf51fhhp-4rkJINxldwSkAophMjgubdHcc5_GPCPUY2KgGEytAkVnNOS
- <https://www.sciencedirect.com/science/article/abs/pii/S0341816220302940>

-

https://www.sciencedirect.com/science/article/pii/S0048969719345449?casa_token=ZiawxlsRQJIAAAA.A:JPjs0ZIIKrN3oxyGs3QJipvwjm3x5rti8xB4bwKODfIDM-4c4c1M8til8xEE8MwOyjbTRs0Rso
- https://esajournals.onlinelibrary.wiley.com/doi/full/10.1890/13-0377.1?casa_token=tdR0smWl6JAAAAA%3A3YcmytQc-S1Oi7v3viR4AN00CgYVVRG3Uw6F-YJxQPzKgCu1le_U7Qbw4ul24bpSi-gYcghYJxEd8NQk).

Here, it is then important to stress the grazing intensity/density factor. From literature it is clear that

reducing grazing intensity increases biodiversity. Once more, please avoid sentences that confuse and may be misinterpreted.

Line 119: what do you mean by “present” richness?

Line 119: replace “or” with “and”. As mentioned above, you cannot tease apart the two factors.

Line 127: replace “intensive” with “extensive”.

Lines from 159: please report on sedaDNA quality.

Lines 245: *Anthyllis vulneraria* is not an alpine taxa (<https://www.infoflora.ch/en/flora/anthyllis-vulneraria.html>) . Please change it starting from tab. ST2.

Lines 246: without the species of *Chaerophyllum*, only the genus is not enough to distinguish between widespread (*C. hirsutum*), mountain (eg. *C. aureum*) or subalpine/alpine (eg. *C. villarsii*) taxa. Please clarify or change accordingly.

Line 258: what do you mean by “low natural landscape”?

Lines 394-396: it is not possible to distinguish such “steep rise” from a sharp increase in DNA quality as compared to older sediments. Please clarify.

Lines 400-405: this paragraph is somehow indicative and representative. It starts with “Richness significantly responds to precipitation” and ends with “Thus, human land use may have been a stronger driver than climate”. Isn’t precipitation part of climate? There is no evidence of human land use influence on richness. Furthermore, it does not tell anything about the nature of such influence. Please revise.

Lines 406-410: Likewise, this paragraph is confusing and misleading. From it, one can easily deduce that erosion is great for biodiversity, which is not clearly the case. Probably authors mean ‘moderate levels of disturbance’.

Lines 433-449: please refer to above-mentioned study highlighting the negative impact of livestock grazing on biodiversity and ecosystem functions.

Lines 454-456: good point. However, you should also consider the broader effects of grazing and especially its intensity. Please revise accordingly.

Lines 460-463: Conclusions are detached from results and literature, but convey a highly misleading message. The problem is not around forest species but simply that species from lower altitudes are more competitive than species from higher altitude. In the context of upward migrating, those more-competitive-lower-altitude species would exclude species from higher altitudes (see eg <https://www.nature.com/articles/nature14952>

<https://www.sciencedirect.com/science/article/pii/S0169534716301306?via=ihub>
<https://www.frontiersin.org/articles/10.3389/fevo.2020.616562/full>)

Lines 642-656: this is key to results reliability and it should be summarize somewhere in the discussion.

Lines 671-679: likewise, please notice that the strong limitations associated with using reads frequency as proxy for abundance (<https://onlinelibrary.wiley.com/doi/10.1111/mec.14920>)

Reviewer #2 (Remarks to the Author):

The manuscript from Dr Garces-Pastor & colleagues “High resolution ancient sedimentary DNA shows that alpine plant biodiversity is a result of human land use” has furtherly improved. With regard to the statistical issues relating to RDA raised by reviewer 1, I can see that the authors tried to apply pWRDA (that should overcome non-linear relationships between dependent and independent variables) having non differences in the results. However, I would be more correct to insert this last analysis in the methods and results.

As written in my previous revisions, although this work does not have such an original framework (i.e. background), it has captured some original aspects (i.e. genetic database, integrated approach, high resolution, number of taxa) of the sedaDNA that can be of interest for a wide audience and could inspire numerous future studies. At this stage, I have no further comments for this manuscript.

Response to reviewers

Dear reviewers,

We thank you very much for your helpful comments, which have contributed to improve the quality of the manuscript.

We have performed the statistical method suggested by the reviewers, namely structural equation models, and our new statistical results coincide with our original conclusions. We have also added further discussion and more references on the effect of grazing on species diversity in the Alps and similar environments.

The authors have addressed all the reviewers' suggestions, which have considerably improved the structure and content of the manuscript, and we respond to them point-by-point below.

Yours,
Sandra Garcés-Pastor and co-authors

Updated figures and tables:
Revised manuscript
Figure: 2, 3, 4, 5, 6
Supplementary Material: SF12, SF13, SF14, SF15
Supplementary Table: ST2, ST15, ST17
Supplementary dataset: S4

Our detailed comments are given below.

Reviewer #1 (Remarks to the Author):

Thanks for taking the review seriously and addressing most concerns. This is a much improved and robust version. As anticipated, new analyses show that climate, and in particular precipitation, plays a major role in explaining biodiversity change and patterns. Furthermore, now it is clear that sedaDNA quality is a major shortcoming, but authors now did a good job in using higher quality data.

Unfortunately, with the GAM model we only know that precipitation and herbivores predict species richness (as indicated in table ST10), but we do not know HOW. That is, we cannot understand whether species richness increases or decreases with precipitation and herbivores, or whether there is a quadratic or exponential relationship among them. This is a crucial point. However, such key information is currently missing. Consequently, all statements around 'species richness increase with herbivores' remain unsupported by results. As such, those claims around livestock grazing lay in the domain of speculations. As already recommended, a gls-type of model would help you in answering such question and solving the problems highlighted previously as well as below.

Our comment: We thank the reviewer for these concerns.

We agree that the shape of the relationships could have been better presented in the original text and we have now made a further effort to improve this. In Fig. 5, the variation of species richness with precipitation and herbivores is shown, which show the direction of the relationship. ST10 table shows the p-values of the GAM analysis, to fulfill the needs of frequentist readers (and reviewers), and Fig. 5 shows the real data, plus the GAM predictions.

We performed a structural equation modelling (SEM) to test direct and indirect effects of climate, wild mammals and domesticates on plant richness with the piecewiseSEM package in R by using the function `gls` on the strict quality dataset. Note that running the same SEM analysis using the function `lm` gives the same results as `gls` for the significant predictors. These analyses offer the possibility to know how and to what extent the variables influence plant richness. One climatic variable (Precipitation) and two human related variables (cow and charcoal) explain the 45% of the plant richness changes through the Holocene. All our all models indicate that both climate and human related factors are the important drivers of plant richness.

Action taken: We performed a structural equation modelling (SEM) and added this information as a new result (pages 7 - 8), expanded the methods section and added Supplementary Table ST11 and Supplementary Figure SF15.

“Furthermore, we used structural equation model (SEM) with the “strict” filtered data to explore how and to what extent the predictors influence on plant richness can be modelled by this approach (Supplementary Table ST17, Fig. 6). The model fit the data reasonably well (Fisher C = 6.259 , df = 18, p-value = 0.995). Results show that plant richness ($R^2 = 0.45$) can be modelled as being significantly and positively affected by precipitation and charcoal. On the other hand, SEM also models precipitation as having an indirect effect on sheep and a negative indirect effect on ibex and chamois, while charcoal has an indirect effect on sheep (Supplementary Table ST17).”

A second unresolved problem is still associated to inferring quantitative data from *sedaDNA*. First, there are strong limitations associated with using reads frequency as proxy for species richness and abundance (see e.g. <https://onlinelibrary.wiley.com/doi/10.1111/mec.14920> and especially the book by Taberlet on eDNA). Second, erosion brings more sediment DNA into the lake, and erosion increases with deforestation and livestock grazing. So, higher estimated taxonomic richness may result from higher erosion and sediment DNA rather than from an actual increase in biodiversity. Unfortunately, all together, these shortcomings impair the reliability of results and solidity of associated claims.

Our comment: We thank the reviewer for his comments.

We are aware of the limitations of using quantitative data from *sedaDNA*. However, we are not using raw “reads frequency” in our analyses. We are introducing a novel relative importance index calculated as a combination of relative read abundance and ratio of positive PCR replicates. Our eight repeated analyses of each sample allows us to do this more conservative quantification than has been done by most studies until now. Even though old literature highlighted the evidently imperfect quantitative mismatch between metabarcoding reads and biomass, recent eDNA metabarcoding papers are showing that semi-quantitative indices give more useful and accurate ecological information than using just presence/absence data (Chen & Ficetola 2020). There are over 300 papers published in the last two years using semi-quantitative metabarcoding approaches.

So there is a growing consensus among molecular ecologists that using a semi-quantitative response, even if it does not perfectly reflect the measured species biomass, provides more useful ecological information than using just binary presence/absence values. We advocate for the use of such semi-quantitative indices, which offer a higher dynamic-range and a more accurate representation of ecological responses by weighting the contributions of different species, instead of a single binary measure.

Regarding the possible bias related to erosion, we agree that this could be an issue. However, a higher erosion would imply an increase of Loss of Ignition (LOI) into the lake. We have tested the relationship between LOI and plant richness with the GAM and SEM analyses, and results do not show a statistically significant relationship between erosion and plant richness. Therefore, an increase of LOI is not related with a higher plant richness.

To overcome such issue, as already mentioned several times, authors can use attribution models, identifying the relative contribution of different factors (climate, surface, ecological, anthropogenic). Those models are commonly done in climate science (see my previous review) and would make results more reliable.

Our comment: Thank you for your continued scrutiny on this issue.

The suggestion to use attribution models to our data is very interesting and we have reexamined the (very interesting and thought-provoking) references suggested by the reviewer. However, the examples of attribution models that were previously suggested by the reviewer for discerning partially-associated factors and properly attributing to humans are unfortunately not applicable to our data because those were examples from Climate Change research based on the multi-analysis of several data sources and simulations from several studies, put together. Unfortunately, a single lake study is not suitable to properly explore such models. As more high-resolution studies become available, it will be possible to perform such models. We have searched for other alternatives and used Structural Equation Models (SEMs). Those models are built using a list of structured equations, which can be specified using most common linear modelling functions in R, and thus can accommodate non-normal distributions, hierarchical structures and different estimation procedures.

We would also like to emphasize that the numerical approaches we now use (RDA, GAM) have been widely applied and tested in the palaeosciences, while SEM is new in palaeoecological *sedaDNA* studies. So for example, even though it is correct that RDA models the relationship between response and variables in a linear way, the method has been extensively tested with datasets which slightly deviate from this assumption (as tested based on DCA 1 gradient lengths in SD units, see pages 7 - 8) and has been shown to perform well even in these situations.

Action taken: In order to perform an additional analysis to reinforce the interpretation of our results for *sedaDNA*-derived plant richness, we used structural equation modelling (SEM) to test direct and indirect effects of climate, wild mammals and domesticates on plant richness. We performed the model by using the function `gls` and the strict quality dataset. Results show that both proxies reflecting climate (precipitation) and human activity (charcoal as a proxy for fire activity, cow DNA) can be successfully modelled as predictors of plant richness. The results therefore agree with our other numerical analysis (RDA, GAM) that indicate that changes in plant diversity in our record coincided with both changes in climate and human activity at Sulsseewli (with some minor differences in the exact results depending on the method used).

Furthermore, authors keep ignoring the evidence that climate explains vegetation patterns. As a matter of facts, when excluding poor DNA data, the association between human land use proxies and vegetation disappears (see lines 217-219 and lines 222-224). Instead, climate always predict vegetation both in poor- and better-quality data as well as both in rda and gam models (see lines 227-229).

Our comment: We thank the reviewer for pointing this out so that we can make this information clearer.

We do not ignore that climate explains part of the vegetation patterns, but we aim to both the human and the climatic influence. As Supplementary Table ST15 shows, the RDA1 of both quality filters shows a significant association between vegetation composition with cow and sheep, indicators of human activity, while precipitation is only significant when using the relaxed filtered data. All the analyses (SEM, RDA, GAM) with both relaxed and strict quality evidence the association between climate and human land use with vegetation composition and plant richness, with an increasing human influence at the end of Holocene.

Action taken: We have rephrased parts of the text to make this clear and incorporated the structural equation modelling. (page 7)

Please adjust the title accordingly by including climate too. Furthermore, given the descriptive, correlative nature of your data, claims about causal relationships should be avoided. This holds true starting from the title, where “is a result of” should be replaced by “is associated with”. The new title shall read like “High resolution ancient sedimentary DNA suggests that regional alpine plant diversity is associated with human land use and climate”.

Our comment: We thank the reviewer for pointing this out

Action taken: Title changed as suggested.

Specific edits:

Lines 66-67 and elsewhere: according to the new analyses, there are no associations between climate and vegetation patterns as of 6 k BP. Please pay attention to stick to your actual results.

Our comment: We thank the reviewer for his comment.

Action taken: the sentence was rephrased to highlight that climate was the only driver of vegetation during the early Holocene. Lines: 67 - 68.

Lines 69: Where are the data showing a “reduced forest cover”? There is no evidence in support of a “reduced forest cover”. Delete.

Our comment: We thank the reviewer for pointing this out so we can make this information clearer.

Action taken: we agree and have changed the abstract text from “Reduced forest cover” to “Associated human deforestation”. Lines: 68 - 69.

Lines 70: “thereby increasing” – once more, please bear in mind that causal processes cannot be discerned with the current analyses. As already mentioned in my previous review, if authors aim at causal inference they should follow a. This is commonly done in climate science where the effects of anthropogenic impact and natural variation are parsed out and attributed to climate change. But such procedure is not unfortunately done in this manuscript. Given the current type of simplistic correlative models (RDA, GAM etc), it is not possible to tease apart the relative contribution of different predictors. Please adjust your language and terminology accordingly and avoid conveying misleading messages.

Our comment: We thank the reviewer for his comment.

Action taken: we added the more sophisticated structural equation modelling and changed the abstract text accordingly. Lines: 68 - 71.

Lines 69–72: These sentences bear many different problems at once. In addition to those just explained above, authors keep pushing their ideological agenda toward deforestation and livestock grazing, ignoring the vast and deep ecosystem damages that those two activities bring. There is a vast literature highlighting the impact of livestock grazing across ecological scales that reduces biodiversity and multiple ecosystem functions. See e.g.:

-
https://onlinelibrary.wiley.com/doi/full/10.1111/ele.13527?casa_token=Tj45v4QN748AAAAA%3A3AD7DpXq_e997jh3lzZPh7jJNKQIdIH3RKv67nNeDEMmBr2DxGC1-aJHgGVcPChdHeYWzu_XkB9KnpW4
- <https://esajournals.onlinelibrary.wiley.com/doi/full/10.1890/ES14-00316.1>
- https://esajournals.onlinelibrary.wiley.com/doi/full/10.1890/15-1234?casa_token=cKyPr3jOcQEAAAAA%3AsJ3BUzyGbl1Vq-j0BPhG7RDx4HnZVPPlg-_LKNLOuXKR_jctDjXOyqoe8ZeT5x5ENs-7iMpOuXt--1g
-
https://onlinelibrary.wiley.com/doi/full/10.1111/rec.12676?casa_token=as9BTd3FWbAAAAA%3ADXsOGK1rKbzuBYmUvmc5p2yhmJK490uPzgF1pVgd5-_VH-eyS-tisO_-1Sm-rZhVEXzQuphOP6Tq19s
-
https://onlinelibrary.wiley.com/doi/full/10.1002/ldr.2668?casa_token=q9IYAf11Zi8AAAAA%3ABmUcnkNIMoXNozW9no35Wchgf51fhhp-4rkJINxldwSkAophMjgubdHcc5_GPCPUY2KgGEytAkVnNOS
- <https://www.sciencedirect.com/science/article/abs/pii/S0341816220302940>
-
https://www.sciencedirect.com/science/article/pii/S0048969719345449?casa_token=ZiawxIsRQJIAAAAA:JPjs0ZIIKrN3oxyGs3QJipvwjm3x5rti8xB4bwK0DfIDM-4c4c1M8tiI8xEE8MwOyjxbTRs0Rso
- https://esajournals.onlinelibrary.wiley.com/doi/full/10.1890/13-0377.1?casa_token=tdR0smWl6JAAAAA%3A3YcmytQc-S10i7v3viR4AN00CgYVRG3Uw6F-YJxQPzKgCu1Ie_U7Qbw4ul24bpSI-gYcghYJxEd8NQk).

Here, it is then important to stress the grazing intensity/density factor. From literature it is clear that reducing grazing intensity increases biodiversity. Once more, please avoid sentences that confuse and may be misinterpreted.

Our comment: We thank the reviewer for this comment and for the recommended references.

Our discussion is based on the evidence provided by our data. We have carefully read the eight references provided by Reviewer 1 and it comes out that five of them (Filazzola et al., 2020; Lindenmayer et al. 2018; Eldridge and Delgado-Baquerizo 2017; García-Ruiz et al. 2020; Yang et al. 2020) deal specifically with animal diversity while the other three (Evans et al., 2015; Eldridge et al., 2015; Hanke et al. 2014) either found no effect (Evans et al., 2015; Eldridge et al., 2015) or support (Hanke et al. 2014) an increase in plant richness (alpha diversity) due to grazing, which is also referred to in Filazzola et al. (2020).

Usually, when precipitation is not a limiting factor like in our case, moderate grazing by large herbivores (hence sheep and cattle) increases richness (Olf & Ritchie, 1998; de Bello & al., 2007; Hanke & al. 2014), in agreement with the "intermediate disturbance hypothesis" originally put forward by Grime in 1978. In addition, an intermediate disturbance like moderate intensity of grazing and/or mowing maintains plant diversity in limiting the competition of dominant species by decreasing their biomass and by allowing the installation of rare species in creating small open spaces. Therefore, it is not just by chance that the highest plant richness found in the world at small spatial grain is found in oligo- to mesotrophic, managed, semi-natural, temperate grasslands like those in the White Carpathians that are managed by grazing and mowing since the Neolithic (Wilson & al., 2012).

Action taken: we changed the sentence in the abstract and introduction to highlight the *low intensity* of agropastoral activities. "Our findings indicate a positive association between low intensity agropastoral activities and precipitation with the maintenance of the unique subalpine and alpine plant diversity of the European Alps." Lines: 71 - 73. We also rephrased a part of the discussion and added some references about the negative impact of grazing on animal diversity (pages 13 - 14).

Line 119: what do you mean by "present" richness?

Our comment: We thank the reviewer for pointing this out

Action taken: we change the word to "current". Line: 122

Line 119: replace "or" with "and". As mentioned above, you cannot tease apart the two factors.

Our comment: We thank the reviewer for his comment.

Action taken: we changed the abstract text accordingly. Line: 123

Line 127: replace "intensive" with "extensive".

Action taken: the word was replaced as reviewer suggested. Line: 132

Lines from 159: please report on sedaDNA quality.

Action taken: We reported as suggested. Line: 170-171

Lines 245: Anthyllis vulneraria is not an alpine taxa (<https://www.infoflora.ch/en/flora/anthyllis-vulneraria.html>). Please change it starting from tab. ST2.

Our comment: We thank the reviewer for his comment. Indeed, *A. vulneraria* s.l. is not a good indicator since there are several subspecies with different elevational distribution from the colline to the alpine (T value = x in Flora indicativa).

Action taken: this species was deleted from the text, the ST2 table and the analyses. We redid the corresponding plots of Figures 2, 3 and 4B, and SF12B, supplementary dataset

Lines 246: without the species of Chaerophyllum, only the genus is not enough to distinguish between widespread (*C. hirsutum*), mountain (eg. *C. aureum*) or subalpine/alpine (eg. *C. villarsii*) taxa. Please clarify or change accordingly.

Our comment: We thank the reviewer for his comment

Action taken: We have specified *Chaerophyllum villarsii*. Line: 261

Line 258: what do you mean by “low natural landscape”?

Our comment: We thank the reviewer for his comment

Action taken: We rephrased the sentence to "suggesting a low mammalian biomass and a natural landscape". Line: 272 - 273

Lines 394-396: it is not possible to distinguish such “steep rise” from a sharp increase in DNA quality as compared to older sediments. Please clarify.

Our comment: We thank the reviewer for his comment

Action taken: We changed the text accordingly. Line: 406

Lines 400-405: this paragraph is somehow indicative and representative. It starts with “Richness significantly responds to precipitation” and ends with “Thus, human land use may have been a stronger driver than climate”. Isn’t precipitation part of climate? There is no evidence of human land use influence on richness. Furthermore, it does not tell anything about the nature of such influence. Please revise.

Our comment: We thank the reviewer for his comment

Action taken: The sentence was deleted from the text.

Lines 406-410: Likewise, this paragraph is confusing and misleading. From it, one can easily deduce that erosion is great for biodiversity, which is not clearly the case. Probably authors mean ‘moderate levels of disturbance’.

Our comment: We thank the reviewer for his comment

Action taken: We agree and adjusted the text accordingly

Lines 433-449: please refer to above-mentioned study highlighting the negative impact of livestock grazing on biodiversity and ecosystem functions.

Our comment: We thank the reviewer for his comment.

The references given by Reviewer 1 do not support that extensive grazing has a negative impact on plant biodiversity, but rather the opposite. In our study, we do not claim that agropastoral activities enhanced the total biodiversity. We consider that it would be counterproductive and misleading to add a citation of an actual study from a very different ecosystem (Australia or Africa) to speculate on what occurred to the past total biodiversity in the Alps. However, we have added a sentence suggesting that more studies should be done and that our results cannot be extrapolated to different ecosystems.

Action taken: we changed part of the paragraph "Our study showcases how late Holocene changes in plant composition and richness around Sulsseewli were associated with increasing human activities, leading to a strongly structured landscape. Based on the intermediate disturbance hypothesis^{61,62}" (page 13).

Lines 454-456: good point. However, you should also consider the broader effects of grazing and especially its intensity. Please revise accordingly.

Our comment: We thank the reviewer for his comment.

Action taken: Already mentioned in the lines before. We add a sentence “Future studies should also consider the impact of grazing on the total biodiversity.”

Lines 460-463: Conclusions are detached from results and literature, but convey a highly misleading message. The problem is not around forest species but simply that species from lower altitudes are more competitive than species from higher altitude. In the context of upward migrating, those more-competitive-lower-altitude species would exclude species from higher altitudes (see eg <https://www.nature.com/articles/nature14952>
<https://www.sciencedirect.com/science/article/pii/S0169534716301306?via=ihub>
<https://www.frontiersin.org/articles/10.3389/fevo.2020.616562/full>)

Our comment: We thank the reviewer for his comment.

Action taken: We changed the paragraph

“In conclusion, the local and regional plant richness of the Subalpine-Alpine landscape in the European Alps is strongly related to millennial agropastoral activities. Currently, high elevation farms and pastures are increasingly abandoned. However, maintaining such activities may be key for preserving the high plant diversity of alpine landscapes that is now threatened by climate change^{17,18,48,63,66}. Intermediate levels of disturbance associated with these extensive land use practices could ensure the continuity of the ecological niches for slow-growing alpine species, and limit the competition that they are facing by the upward shift of more competitive species from lower elevation^{3,4,10,71}“ Lines: 483 - 490.

Lines 642-656: this is key to results reliability and it should be summarize somewhere in the discussion.

Our comment: We thank the reviewer for his comment.

Our results are robust to whether some lower-quality samples are included or not. Given the length of the manuscript we consider that providing this information also in the discussion would be repetitive.

Action taken: we added a sentence about the DNA quality in the results (page 5).

Lines 671-679: likewise, please notice that the strong limitations associated with using reads frequency as proxy for abundance (<https://onlinelibrary.wiley.com/doi/10.1111/mec.14920>)

Our comment: We thank the reviewer for his comment.

As explained above, we propose the use of a semi-quantitative relative importance index, which offers a higher dynamic-range and a more accurate representation of ecological responses than a single binary measure, by weighting the contributions of different species. Even though methods for abundance estimation have major limitations, they can provide more useful information on temporal variation of ecosystem functions, and this is the growing consensus among eDNA and *seada*DNA researchers (Chen & Ficetola 2020 <https://doi.org/10.1002/edn3.79>).

Reviewer #2 (Remarks to the Author):

The manuscript from Dr Garces-Pastor & colleagues “High resolution ancient sedimentary DNA shows that alpine plant biodiversity is a result of human land use” has furtherly improved. With regard to the statistical issues relating to RDA raised by reviewer 1, I can see that the authors tried to apply pWRDA (that should overcome non-linear relationships between dependent and independent variables) having non differences in the results. However, I would be more correct to insert this last analysis in the methods and results.

As written in my previous revisions, although this work does not have such an original framework (i.e. background), it has captured some original aspects (i.e. genetic database, integrated approach, high resolution, number of taxa) of the *seada*DNA that can be of interest for a wide audience and could inspire numerous future studies. At this stage, I have no further comments for this manuscript.

Our comment: We thank the reviewer very much for his comment along the reviewing process.

Action taken: We added the graphs of pWRDA in the Supplementary material (Supplementary Figure SF13) and we adjusted the text in the statistics part of the Methods: In addition, we performed a piecewise redundancy analysis¹⁰¹ to crosscheck the robustness of the RDA analyses of the effect of the 12 explanatory variables on the species composition (Supplementary Figure SF12A).” and in the results section “We also explored piecewise RDA, however the results did not differ from those of classical RDA (see Supplementary Fig. SF13)”.

REVIEWERS' COMMENTS

Reviewer #1 (Remarks to the Author):

Thanks for addressing all my suggestions and concerns.
The manuscript is solid and in a great shape now.
It is going to make a nice impact.